# Cotranslational N-degron masking by acetylation promotes proteome stability in plants

Eric Linster[1], Francy L. Forero Ruiz[1], Pavlina Miklankova[1], Thomas Ruppert[2], Johannes Mueller[3], Laura Armbruster[1], Xiaodi Gong[1], Giovanna Serino [4], Matthias Mann [3], Rüdiger Hell[1] & Markus Wirtz [1✉]

N-terminal protein acetylation (NTA) is a prevalent protein modification essential for viability in animals and plants. The dominant executor of NTA is the ribosome tethered N$^\alpha$-acetyl-transferase A (NatA) complex. However, the impact of NatA on protein fate is still enigmatic. Here, we demonstrate that depletion of NatA activity leads to a 4-fold increase in global protein turnover via the ubiquitin-proteasome system in Arabidopsis. Surprisingly, a concomitant increase in translation, actioned via enhanced Target-of-Rapamycin activity, is also observed, implying that defective NTA triggers feedback mechanisms to maintain steady-state protein abundance. Quantitative analysis of the proteome, the translatome, and the ubiquitome reveals that NatA substrates account for the bulk of this enhanced turnover. A targeted analysis of NatA substrate stability uncovers that NTA absence triggers protein destabilization via a previously undescribed and widely conserved nonAc/N-degron in plants. Hence, the imprinting of the proteome with acetylation marks is essential for coordinating proteome stability.

[1] Centre for Organismal Studies Heidelberg, Heidelberg University, Heidelberg, Germany. [2] Center for Molecular Biology Heidelberg, Heidelberg University, Heidelberg, Germany. [3] Max-Planck-Institute for Biochemistry, Martinsried, Germany. [4] Department of Biology and Biotechnology, Sapienza Università di Roma, Rome, Italy. ✉email: markus.wirtz@cos.uni-heidelberg.de

As sessile organisms, plants have to fight environmental challenges on site. The proteome's dynamic plasticity is one of the most critical mechanisms allowing plants to acclimate to changes in their environment rapidly. This notion is supported by the substantially elaborated ubiquitin-proteasome system (UPS) in plants compared to humans[1]. Despite the essential importance of protein degradation, we are only now beginning to understand how plants control proteostasis upon stress and under favorable growth conditions. Protein modifications have been identified as crucial determinants of protein stability in eukaryotes and are highly regulated upon diverse plant stress conditions. One of the most pervasive protein modifications is N-terminal protein acetylation (NTA). NTA occurs on 80-90% of human and Arabidopsis soluble proteins and is executed by up to five ribosome-associated N-terminal acetyltransferases (Nat) complexes, of which NatA, NatB and NatC are conserved in all eukaryotes[2]. Disturbance of NTA in humans causes fatal diseases like Ogden syndrome, whilst enhanced NTA is associated with deregulated cell proliferation in specific cancer types[3,4].

The NatA complex consists of the catalytically active subunit NAA10 and the ribosome-anchoring subunit NAA15, and targets nascent chains of proteins after the initiator methionine (iMet) is cleaved by methionine aminopeptidase (MetAP). Consequently, a canonical NatA substrate is defined by cotranslational iMet removal resulting in a nascent chain displaying an amino acid that can be acetylated by NAA10[5]. In Arabidopsis and humans, 40% of proteins are subjected to this N-terminal protein trimming by NatA. The remaining 40–50% of N-terminally acetylated proteins are acetylated predominantly at the iMet by the NatB, NatC, NatE or NatF complex. In plants, Nat complexes are particularly important for the resistance towards diverse abiotic and biotic environmental stresses[6–10]. The dynamic regulation of the NatA abundance by the phytohormone abscisic acid (ABA) is essential for drought stress resilience. However, the NatA-dependent mechanism for the regulation of drought stress responses remains to be determined. Only in a few cases NTA has been reported to affect protein functionality[2,11]. Thus, controlling the activity of individual proteins is unlikely to explain the pervasive NTA of bulk proteins[12].

In yeast and humans, NTA can create N-degrons recognized by the Ac/N-degron pathway and leading to the destruction of proteins by the UPS. On the contrary, another set of proteins was stabilized by NTA[13,14]. Taken together a direct impact of NTA on protein stability has been documented for less than 30 proteins in all eukaryotic model species. Thus the impact of NTA on global proteome stability remains unclear in eukaryotes[9,13–18]. However, a recent study demonstrated an overlap of substrates recognized by NatA and the IAP-type E3-ubiquitin ligases in vitro, suggesting that N-terminal acetylation is relevant for protein stabilization in metazoans[19].

Here we show that impairment of NatA-dependent NTA results in a global destabilization of the proteome in Arabidopsis and discover a novel degron that marks the majority of non-acetylated cytosolic proteins for degradation via the ubiquitin system.

## Results

**NatA depletion caused enhanced protein degradation.** Since loss of NatA causes embryo-lethality in plants[10], we independently down-regulated both subunits of the NatA complex by an amiRNAi-approach and tested the global protein degradation rates in leaves after feeding of isotope-labeled amino acids. The depletion of the NatA complex substantially enhanced the protein degradation rate, causing up to 4-fold faster protein destruction

(Fig. 1a), when the NAA10 abundance was decreased to 25% or 20% of wild type level in amiNAA10 lines 18 or 23, respectively[10]. The combined activity of metallo-, serine-, acid-, and sulfhydryl-type proteases was not enhanced in any of the NatA-depleted plants (Supplementary Fig. 1). However, the NatA-depletion triggered a specific increase of the proteasome activity (Fig. 1b), which negatively correlated with the previously demonstrated decreased growth of the individual amiRNAi lines with depleted NatA levels[10]. The finding of increased proteasome activity in NatA depleted plants was supported by the immunological detection of the accumulation of the lid and the core subunit of the 26 S proteasome RPN10 and PBA1, respectively, in amiNAA10 (Supplementary Fig. 2a–d). The accumulation of both subunits was not triggered by enhanced steady state transcript levels (Supplementary Fig. 2e), albeit, the pathways of ubiquitin-mediated proteolysis is transcriptionally induced upon NatA depletion[10]. The endogenous ubiquitination rate also increased in NatA depleted plants and resulted in most significant accumulation of poly-ubiquitinated proteins in the transgenic line with the most substantial depletion of NatA activity. In line with these observations, all NatA depleted plants accumulated higher amounts of ubiquitinated proteins than the wild type after pharmacological inhibition of the proteasome (Fig. 1c, Supplementary Fig. 2f). Furthermore, enhanced neddylation of Cullin 1 demonstrated that Cullin-RING E3 ligases (CRLs[20]), contributed to the enhanced in vivo ubiquitination activity in NatA depleted plants (Fig. 1d, Supplementary Fig. 2g). Our data strongly suggests that NatA depletion caused faster protein degradation by enhancing the endogenous ubiqitination rate and increasing the capacity of the proteasome as shown by enhanced abundance and activity of the proteasome in soluble protein extracts.

**The majority of degraded proteins in NatA depleted plants were NatA substrates.** Next, we aimed to identify the proteins that were destroyed by the UPS when NatA was depleted. Inhibition of the proteasome by MG132 followed by affinity enrichment of ubiquitinated proteins with the UbiQapture-Q matrix resulted in 1.6-fold more bona fide poly-ubiquitinated proteins captured in NatA depleted plants when compared to wild type as detected with a ubiquitin-specific antiserum (Fig. 1e). This increase in bona fide polyubiquinitated proteins correlated well with the increase of total protein after Ubi-Qapture enrichment from amiNAA10 plants (1.5-fold increase, $p < 0.05$, Fig. 1f). Out of the 232 identified proteins that were significantly enriched by the Ubi-Qapture matrix in NatA depleted plants 162 (70%) were canonical NatA substrates (Supplementary Data 1 and 2, Fig. 1g), implying a significant enrichment of NatA substrates in the fraction of the poly-ubiquitinated proteins (Fisher´s exact test, $p$-value $< 0.0001$). A gene ontology enrichment analysis revealed that poly-ubiquitinated proteins in NatA depleted plants were predominantly cytosolic proteins (141 of 232) and associated with the responses to diverse stresses and protein-folding (Supplementary Data 3).

**Cytosolic NatA substrates are destabilized in NatA depleted plants.** Despite the induction of the UPS for degradation of NatA substrates, the total protein content was apparently not affected in leaves of NatA depleted plants (Fig. 2a, Supplementary Fig. 3a). A quantitative analysis of the steady-state protein levels by shotgun mass spectrometry uncovered that the abundance of only 92 out of the 1.238 detected proteins was significantly affected in plants depleted for the catalytic subunit of NatA (Fig. 2b, Supplementary Data 4). However, 83% of the proteins that displayed lowered steady-state levels in NatA depleted plants were canonical NatA substrates (Supplementary Data 5, permutation-based FDR ≤ 1%,

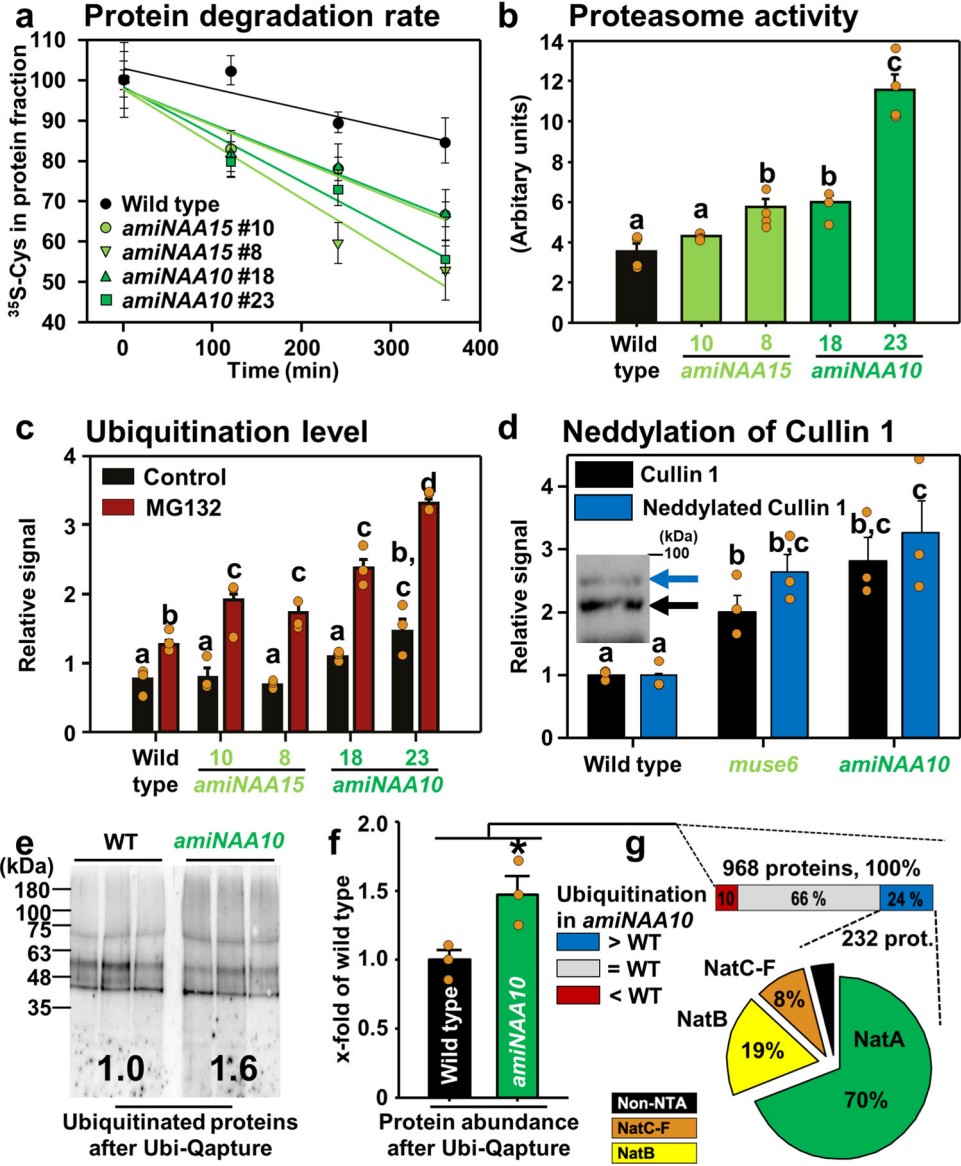

**Fig. 1 Depletion of NatA activity causes enhanced degradation of proteins via the ubiquitin-proteasome system. a** Time-resolved analysis of protein degradation rate in leaves of wild type (black) and four individual lines depleted for the ribosome anchoring (NAA15, light green) or the catalytically active subunit (NAA10, dark green) of the NatA complex. $n = 4$ biologically independent samples, except for WT 360 min, *amiNaa10* # 18 360 min, *amiNAA10* #23 120 min, 240 min, amiNaa15 #8 120 min, 360 min, and *amiNaa15* #10 360 min $n = 3$. **b** Proteasome activity in wild type and NatA activity depleted plants. Different letters indicate individual groups identified by pairwise multiple comparisons with a Holm-Sidak, one-way ANOVA ($p < 0.05$, $n = 4$ biologically independent samples, except for *amiNaa10* #18 $n = 3$). **c** Relative level of poly-ubiquitinated proteins as determined with the ubiquitin-specific antiserum (α-UBQ11; Agrisera) in leaves of wild type and NatA depleted plants in the presence (red) or absence (black) of the proteasome inhibitor MG132. Different letters indicate individual groups identified by pairwise multiple comparisons with a Holm-Sidak, one-way ANOVA ($p < 0.05$, $n = 3$ biologically independent samples). **d** Abundance and activation status of Cullin-RING E3 ligase (CRL) complexes as determined by neddylation of Cullin isoform 1 in wild type and transgenic lines depleted of NAA15 (*muse6*) and NAA10 (*amiNAA10*). Different letters indicate individual groups identified by pairwise multiple comparisons with a Holm-Sidak, one-way ANOVA ($p < 0.05$, $n = 3$ biologically independent samples) **e** Immunological detection of poly-ubiquitinated proteins using an ubiquitin-specific antibody (α-UBQ11; Agrisera) in the wild type and *amiNAA10* line 23 (*amiNAA10*) after selective enrichment with the Ubi-Qapture Q™ matrix. **f** Protein amount of affinity enriched bona fide poly-ubiquitinated proteins. Asterix indicate individual groups identified by pairwise multiple comparisons with a Holm-Sidak, one-way ANOVA ($p < 0.05$, $n = 3$ biologically independent samples) **g** Quantitative proteomics of bona fide poly-ubiquitinated proteins revealed that 24% of quantified proteins (232 proteins) were significantly (student t-test, $p < 0.05$) more ubiquitinated in *amiNAA10* (>1.5-fold more than wild type) or either only found in all *amiNAA10* replicates ($n = 3$ biologically independent samples). The pie chart depicts the classification of Nat substrates in the fraction of significantly more ubiquitinated proteins in *amiNAA10* plants. Data are shown as mean ± standard error. Immunological detections of proteins quantified in **c**, **d** were independently repeated and blots are shown in the source data file.

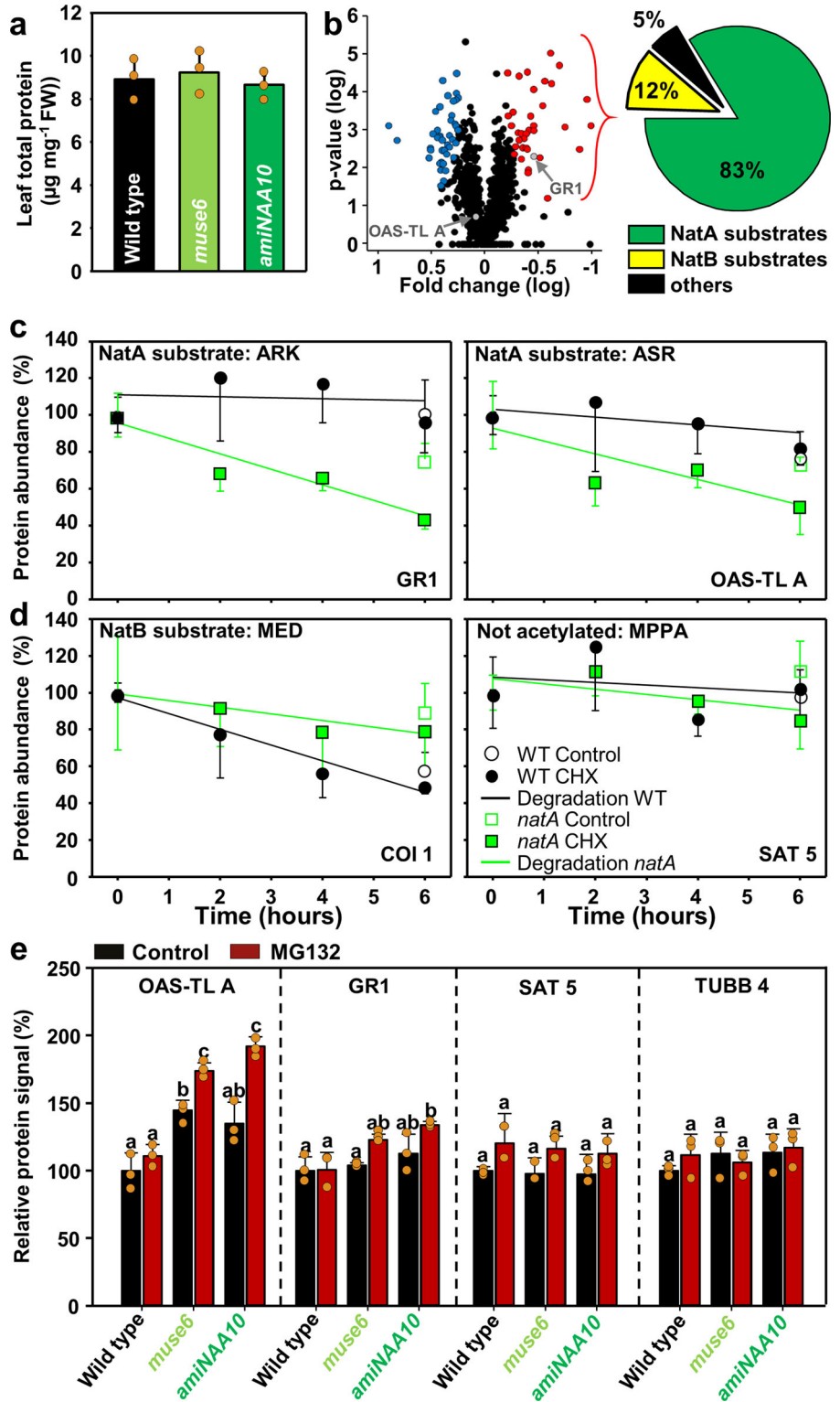

Fisher´s exact test, p-value < 0.0001 for the enrichment of NatA substrates in the fraction of low abundant proteins in *ami-NAA10*). We selected the decreased protein glutathione reductase 1 (termed [ARK]GR1 due to excision of the iMet, AT3G24170, −2.8-fold, $p < 0.05$) and the not-significantly accumulated protein O-acetylserine(thiol)lyase A ([ASR]OAS-TL A, AT4G14880, 1.2-fold) for time-resolved destabilization assays since both cytosolic proteins are canonical NatA substrates. Prior to this analysis, the steady-state protein levels of [ASR]OAS-TLA and [ARK]GR1 in NatA depleted plants were independently confirmed by immunological detection with specific antisera (Supplementary Fig. 3b, c). The cycloheximide chase assays for [ARK]GR1 and [ASR]OAS-TL A demonstrated significantly enhanced degradation of both NatA substrates in NatA depleted plants (Fig. 2c, Supplementary Fig. 3d). In contrast, cytosolic proteins that are not recognized as substrates and thus not acetylated by NatA, like the OAS-TL A interacting protein [MPP]SAT5 (Serine-Acetyl-Transferase 5, AT1G55920) and [MED]COI1 (COronatine-Insensitive protein 1,

**Fig. 2 Depletion of NatA activity does not affect total protein steady-state level but significantly destabilizes selected NatA substrates.**
**a** Concentration of total proteins in leaves of wild type and NatA depleted plants *muse6* and *amiNAA10*, (n = 3 biologically independent samples).
**b** Comparison of leaf proteins in wild type and *amiNAA10* as volcano plot to identify changes in the leaf-proteome due to depletion of the catalytic NatA subunit (*amiNAA10* versus wild type). Significantly altered proteins in *amiNAA10* are labeled in color (red, decreased, blue, accumulated, FDR < 0.01, n = 4 biologically independent samples). The pie diagram displays the classification of Nat substrates in the fraction of significantly decreased proteins in *amiNAA10*. **c**, **d** Time-resolved degradation analysis of selected NatA substrates (**c** GR1, OAS-TL A) and proteins that are not N-terminally acetylated by NatA (**d** COI1, SAT5) in the wild type (circle) and *muse6* carrying a point mutation causing lowered NatA activity[9], box) in the presence (filled) or absence (control, empty) of the translation inhibitor cycloheximide (CHX, n = 4 biologically independent samples). **e** Relative level of OAS-TL A, GR1, SAT5, and TUBB4 proteins as determined with the specific antisera in leaves of wild type and NatA depleted plants in the presence (red) or absence (black) of the proteasome inhibitor MG132. Data represent mean ± standard deviation. Different letters indicate individual groups identified by pairwise multiple comparisons with a Holm-Sidak, one-way ANOVA (p < 0.05, n = 3 biologically independent samples). Immunological detections of proteins quantified in **c**, **d** and **e** were independently repeated and blots are shown in the source data file.

AT2G39940) were not destabilized in NatA depleted plants (Fig. 2d, Supplementary Fig. 3d). Since the steady-state level of the destabilized ASROAS-TL A was unaffected by NatA depletion, we tested the accumulation of ASROAS-TL A after proteasome inhibition by MG132. Short-term inhibition of the proteasome resulted in significantly faster accumulation of ASROAS-TL A in NatA depleted plants when compared to wild type (Fig. 2e, Supplementary Fig. 4), suggesting that the unaffected steady-state levels of the destabilized ASROAS-TL A were a result of enhanced ASROAS-TL A translation in NatA depleted plants. Importantly, enhanced translation was not observed for ARKGR1, and proteins that are not recognized by NatA (MPPSAT5 and MRETUBB4, tubulin β4).

**Enhanced protein degradation is counteracted by increased translation**. Since 40% of the proteome is acetylated by NatA, we hypothesized that translation must be significantly upregulated in NatA depleted plants to maintain the steady-state proteome level. It should be noted in this context that the costs for translation can reach up to 38% of total cellular ATP consumption in wild type Arabidopsis leaves[21], and that protein turnover is known to negatively correlate with the growth rate in the diverse Arabidopsis accessions[22]. Despite the substantial costs of translation in wild type plants, incorporation of isotope-labeled 35S-Met and 35S-Cys into proteins increased up to 4-fold in leaves of NatA depleted plants (Fig. 3a). This higher global translation rate was due to the selective enhancement of translation for diverse proteins (Fig. 3b). In plants, the sensor kinase Target of Rapamycin (TOR) is a critical regulator of the ribosome amount due to phosphorylation of the kinase S6K (Small-ribosome subunit 6 Kinase)[23] and the translation efficiency of stress-related genes due to phosphorylation of the translation initiation factor eIF3h[24]. NatA depletion triggered the increase of TOR activity (Supplementary Fig. 5), resulting in up to 4-fold higher phosphorylation of S6K at T449 and, consequently, a significant accumulation of rRNA (Fig. 3c, d). We applied time-resolved biorthogonal non-canonical amino acid tagging (BONCAT) to identify the more efficiently translated proteins in NatA depleted plants after selective enrichment of the translatome[25]. The incorporation of the trackable Met-analog azidohomoalanine was linear during the period of the analysis and independently confirmed the higher translation rate in NatA depleted plants (Fig. 3e, f). Quantitative proteomics of the newly translated proteins (913 proteins detected, Supplementary Data 6) in the wild type and NatA depleted plants uncovered that 45% of identified proteins were more translated upon NatA depletion (Supplementary Data 7). The vast majority of these proteins were NatA substrates (72%, Fisher´s exact test for enrichment of NatA Substrates, p-value < 0.0001, Fig. 3g). Comparison of the ubiquitome and the translatome of NatA depleted plants revealed that 65 proteins (30% of the NatA

depletion-induced ubiquitome alteration) were more ubiquitinated and more translated. 72% of the proteins with enhanced turnover in NatA depleted plants were canonical NatA substrates (Supplementary Data 8, Fisher´s exact test for enrichment of NatA Substrates, p-value < 0.0001), suggesting that selective destabilization of NatA substrates due to impaired N-terminal acetylation was counteracted by their enhanced translation to maintain their steady-state level in the mutant lines (Fig. 2b). In support of this hypothesis, we found the translation of the OAS-TL A protein to be significantly enhanced (1.7-fold, p = 0.02), while GR1 translation was unaffected by NatA depletion (Supplementary Data 6), explaining the difference in the steady-state levels of the two destabilized NatA substrates (Fig. 2c).

**Identification of nonAc-X²/N-degron containing NatA substrates *in planta***. To provide direct evidence for the enhanced protein turnover by decreased NTA of NatA substrates, we assessed OAS-TL A turnover using the tandem-Fluorescent timer system, which allows for non-invasive quantification of relative protein half-lifetime in plants[26]. OAS-TL A's relative half-lifetime with native N-terminus (ASROAS-TL) was significantly lower in NatA depleted plants compared to wild type (Fig. 4a). In contrast, the relative half-lifetime of proteins that are not targeted by NatA (e.g., tubulin β4 (MRETUBB4) or SAT5 (MPPSAT5)) was unaffected in NatA depleted plants (Fig. 4b, c). A proline residue at position 3 is known to inhibit substrate recognition by the NatA complex in diverse metazoa[27] and plants[10]. To ultimately prove that the absence of NTA of alanine in position 2 (Ala2) is causing the destabilization of the NatA substrate ASROAS-TL A, we genetically engineered an OAS-TL A protein mutant with impaired NTA by inserting a proline at position 3 (APSOASTL A). The APSOASTL was significantly destabilized in the wild type and displayed a similar relative protein half-life time to that of the native ASROAS-TL in NatA depleted plants. Remarkably, APSOASTL was not further destabilized in NatA depleted plants, providing ultimate evidence that the absent NTA of Ala2 is responsible for the destabilization of OAS-TL A in plants (Fig. 4a). Based on these findings, we named this novel destabilizing signal nonAc-X²/N-degron (X² = Ala). To judge the generality of nonAc-X²/N-degron-induced destabilization, we randomly selected ten cytosolic NatA substrate candidates and tested the impact of NTA on their stability by applying the same strategy (Supplementary Table 1). The candidates' identity as proper NatA substrates and the inhibitory impact of proline at position 3 for recognition by the NatA in the three tested NatA substrate groups was verified (X² = Ala: group 1, Gly: group 2 or Ser: group 3, Supplementary Fig. 6). Eight out of the ten NatA substrates were significantly destabilized by inhibiting NTA at position 2. This destabilization also occurred when Ser or Gly occupied position 2 (Fig. 4d). Seven of these NatA substrates

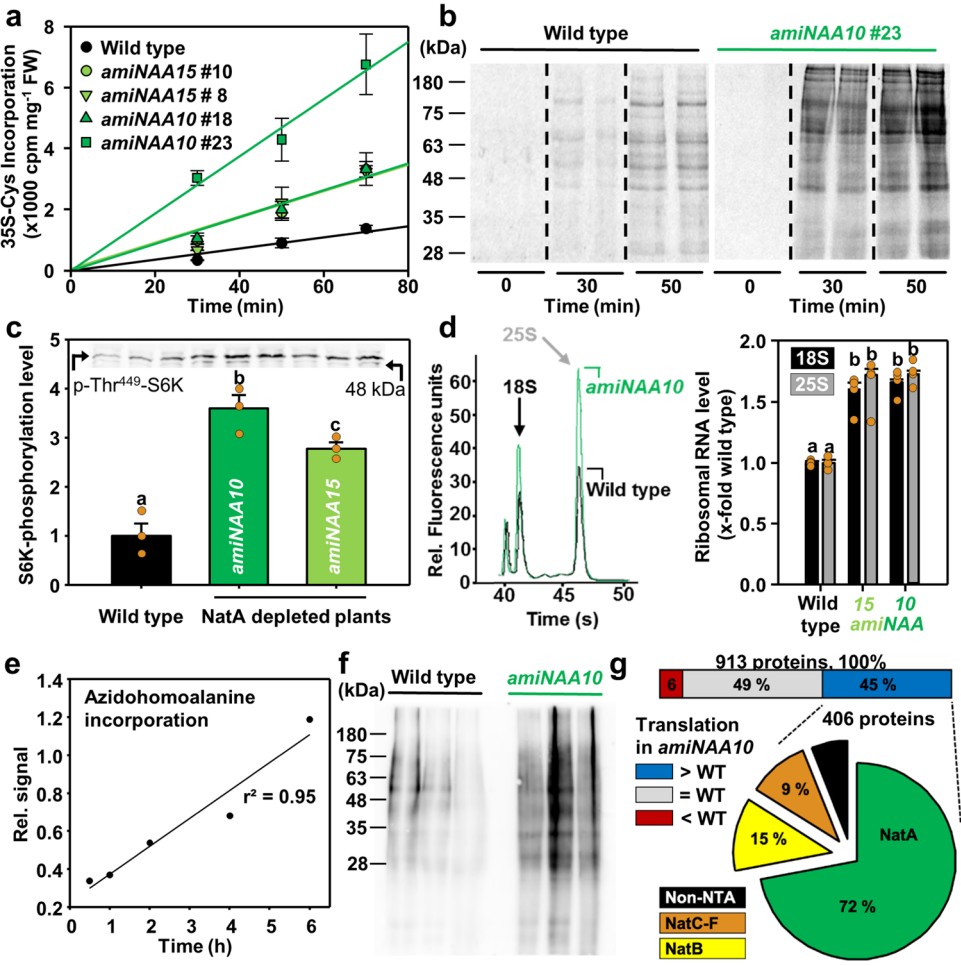

**Fig. 3 NatA depleted plants display higher translation rates of NatA substrates that are facilitated by TOR-induced production of ribosomes. a** Time-resolved incorporation of isotope-labeled sulfur amino acids into foliar proteins of the wild type and four NatA depleted lines (*amiNAA10*, *amiNAA15*). $n = 4$ biologically independent samples, except for WT 70 min, *amiNaa10* #18 70 min, *amiNaa10* #23 50 min, 70 min, *amiNaa15* #8 70 min, and *amiNaa15* #10 70 min $n = 3$. Data are shown as mean ± standard deviation. **b** Auto-radiogram of SDS-PAGE separated foliar proteins from wild type and *amiNAA10* after incorporation of isotope-labeled sulfur-amino acids for indicated time points ($n = 2$ biologically independent samples). **c** Phosphorylation of T449 of S6K by the sensor kinase Target of Rapamycin in leaves of wild type and NatA depleted plants as determined with a phospo-specific antiserum. Different letters indicate individual groups identified by pairwise multiple comparisons with a Holm-Sidak, one-way ANOVA ($p < 0.05$, $n = 3$ biologically independent samples). Data are shown as mean ± standard error. **d** Quantification of 18 S and 25 S ribosomal-RNAs in leaves of wild type and NatA depleted plants. Different letters indicate individual groups identified by pairwise multiple comparisons with a Holm-Sidak, one-way ANOVA ($p < 0.0.5$, $n = 4$ biologically independent samples). Data are shown as mean ± standard deviation. **e** Verification of linear azidohomoalanine incorporation into proteins derived from leaves of wild type for indicated time points ($n = 2$ biologically independent samples). **f** Comparison of azidohomoalanine incorporation for three hours into foliar proteins of wild type and NatA depleted plants after azidohomoalanine-mediated biotin labeling ($n = 3$ biologically independent samples).
**g** Proteomic analysis of newly translated proteins after selective enrichment in leaves of wild type and *amiNAA10* plants. The pie diagram depicts the classification of Nat substrates in the fraction of more efficiently translated proteins in *amiNAA10* plants.

were found to be destabilized in NatA depleted plants, and one NatA substrate, NHO1, could not be detected in the cytosol of NatA depleted plants. This demonstrated that the absence of NTA was the causal effect for decreased relative half-lifetime of these proteins (Fig. 4d, Supplementary Fig. 7). We could further show that the destabilizing effect of NTA inhibition at position 2 was not restricted to highly stable NatA substrates (Supplementary Fig. 8). In two cases, CYP19 and UGE1, NTA inhibition at position 2 had no impact on protein stability in the wild-type Arabidopsis plants. In agreement, CYP19 and UGE1 were also not destabilized in NatA depleted plants. These findings support the notion that nonAc-X²/N-degron functionality requires additional parameters encoded in appropriately positioned domains downstream of the N-terminus, e.g., surface-exposed Lys-residues for ubiquitination[28] or accessibility of the N-terminus for recognition[17].

## Discussion

Since NatA imprints 40% of the proteome in plants, we suggest that absent masking of the nonAc-X²/N-degron in many NatA substrates substantially contributes to the observed higher protein turnover in NatA depleted plants. In agreement with this notion, the enhanced proteome turnover was predominantly based on the degradation of NatA substrates, and N-terminally acetylated NatA substrates were significantly overrepresented in the fraction of stable abundant proteins in plants[29]. Since the depletion of the ribosome-anchoring subunit NAA15 resulted in an increase of proteome turnover comparable to that caused by depletion of the catalytically active subunit NAA10, we suggest that cotranslational NTA is required to chemically block the N-terminus to ensure stabilization of the nascent polypeptide and prevent its unwanted turnover. A similar destabilizing effect was previously shown in humans for the non-acetylated ^MQRGS2 protein that

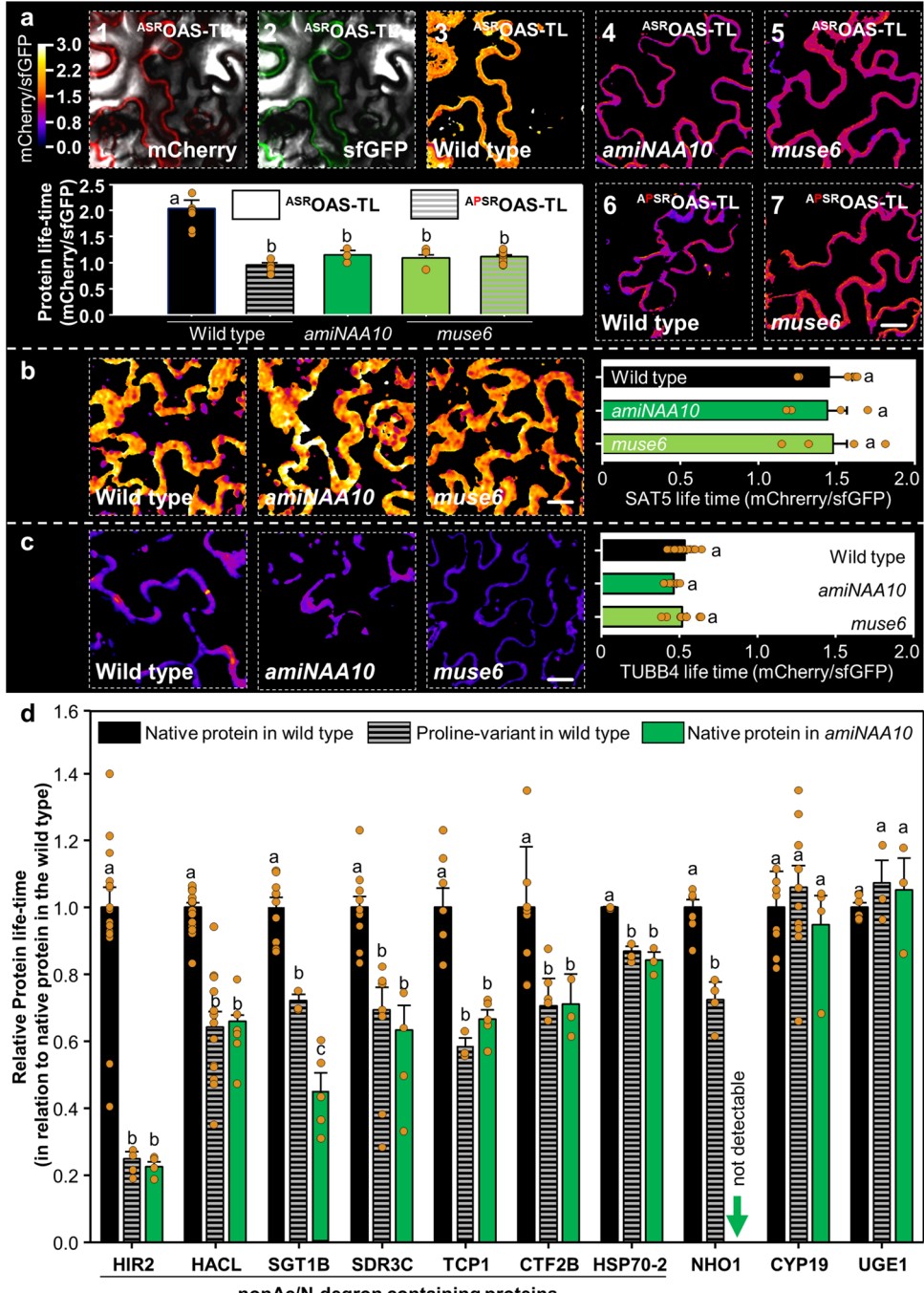

was degraded by the Arg/N-degron pathway. Surprisingly, acetylated $^{MQ}$RGS2 was recognized by the Ac/N-degron pathway, demonstrating that NTA of the iMet can redirect proteins between different branches of the N-degron pathway system[30]. In contrast to other protein modifications, e.g., ubiquitination or Lys-ε-acetylation, NTA is irreversible[11], implying that the stability of many NatA substrates is intrinsically determined at the moment when these proteins are synthesized. However, co-translationally imprinted N-degrons can also contribute to conditional protein quality control when they are unshielded upon stress-induced protein misfolding or exposed in subunits of multi-protein complexes produced in non-stoichiometric amounts[17,31]. As a result of its cotranslational mode and its irreversibility, NTA has been viewed as static in the past[11]. This view on NTA is obsolete in plants for two reasons: first, NTA is

rapidly regulated upon environmental stimuli by the phytohormone-system[10] and second, a significant fraction of NatA substrates is only partially N-terminally acetylated[6–8,10], implying additional regulatory mechanisms controlling the activity of NatA on the nascent chains extruding from the ribosome exit tunnel. Furthermore, crystallization of the trimeric metazoan NatA-HYPK complex uncovered a structural basis for NatA regulation by its binding partner HYPK in vitro[32,33]. The recent identification of E3 ubiquitin ligases (N-recognins, named "inhibitor of apoptosis proteins" (IAPs), specifically recognizing non-acetylated NatA substrates, provides evidence for the existence of this novel N-degron pathway in humans[15,19,34]. If the human nonAc-X²/N-degron pathway is as ubiquitous as in plants is unclear[35]. In humans, IAPs bind unacetylated N-termini of NatA substrates that mimic the IAP binding motif (IBM) present

**Fig. 4 Non-invasive in vivo determination of relative protein half-lifetimes in NatA depleted plants. a** Quantification of relative protein half-lifetimes with the tandem-Fluorescence Timer (tFT) is based on the different maturation times of the fluorescent mCherry (a1) and the super-folding green fluorescent protein (sfGFP, a2) encoded on the same polypeptide chain in fusion with the protein of interest (wild type OAS-TLA, $^{ASR}$OAS-TL-tFT). The mCherry/sfGFP signal ratio (a3, black bar) is a direct readout for the age of the polypeptide chain pool in the cytosol of transiently transformed epidermal leaf cells (cell 1) and positively correlates with the stability of the POI-tFT. Expression of the NatA substrate $^{ASR}$OAS-TL-tFT in transgenic plants depleted for the catalytic (amiNAA10, a4, dark green) or the ribosome anchoring subunit of NatA (muse6, a5, light green) resulted in significant lower relative $^{ASR}$OAS-TL-tFT protein half-lifetime. Inhibition of NTA of OAS-TL by the introduction of a proline at position 3 ($^{APSR}$OAS-TL-tFT, red shaded) decreased the relative protein half-lifetime in the wild type (a6) but did not further destabilize the protein in the NatA mutant muse6 (a7). ($p < 0.05$, WT $^{ASR}$OAS-TL-tFT $n = 8$, WT $^{APSR}$OAS-TL-tFT $n = 6$, amiNaa10 $^{ASR}$OAS-TL-tFT $n = 4$, muse6 $^{ASR}$OAS-TL-tFT $n = 4$ and muse6 $^{APSR}$OAS-TL-tFT $n = 11$). Data are shown as mean ± standard error **b, c** The relative protein half-lifetime of the non-NatA substrates $^{MPP}$SAT5-tFT (WT $^{MPP}$SAT5-tFT $n = 5$, amiNaa10 $^{MPP}$SAT5-tFT $n = 4$, and muse6 $^{MPP}$SAT5-tFT $n = 4$). **b** and $^{MRI}$TUBB4-tFT (WT $^{MRI}$TUBB4-tFT $n = 20$, amiNaa10 $^{MRI}$TUBB4-tFT $n = 6$, and muse6 $^{MRI}$TUBB4-tFT $n = 8$) **c** was not affected by NatA depletion. Scale bar, 15 μm Data are shown as mean ± standard error **d**, Relative protein lifetime of ten cytosolic NatA substrates in wild type (black) and amiNAA10 (green). Definition as canonical nonAc-X²/N-degron containing protein is based on destabilization of the protein by absent NTA due to protein engineering in the wild type (red shaded) or expression of the native protein in NatA depleted plants (green, amiNAA10). The NHO1-tFT protein abundance was below the detection limit in the cytosol of amiNAA10 and muse6 (Supplementary Fig. 7h). Different letters indicate individual groups identified by pairwise multiple comparisons with a Holm-Sidak, one-way ANOVA ($p < 0.05$, WT native HIR2 $n = 16$, WT proline HIR2 $n = 4$, amiNaa10 native HIR2 $n = 4$, WT native HACL $n = 19$, WT proline HACL $n = 13$, amiNaa10 native HACL $n = 6$, WT native SGT1B $n = 11$, WT proline SGT1B $n = 3$, amiNaa10 native SGT1B $n = 5$, WT native SDR3 $n = 12$, WT proline SDR3 $n = 7$, amiNaa10 native SDR3 $n = 4$, WT native TCP1 $n = 8$, WT proline TCP1 $n = 3$, amiNaa10 native TCP1 $n = 5$, WT native CTF2B $n = 9$, WT proline CTF2B $n = 5$, amiNaa10 native CTF2B $n = 3$, WT native HSP70-3 $n = 15$, WT proline HSP70-2 $n = 3$, amiNaa10 native HSP70-2 $n = 3$, WT native NHO1 n $= 8$, WT proline NHO1 $n = 4$, WT native CYP19 $n = 10$, WT proline CYP19 $n = 10$, amiNaa10 native CYP19 $n = 4$, WT native UGE1 $n = 6$, WT proline UGE1 $n = 3$, and amiNaa10 native UGE1 $n = 3$). Data are shown as mean ± standard deviation.

in mitochondria-localized proteins displaying an AVPX at their mature N-terminus (like DIABLO and HRT2)[19]. Since these non-acetylated N-termini are created inside the mitochondria by transit peptide removal, these proteins can only interact with IAPs upon mitochondrial leakage, which is a known inducer of apoptosis in humans. In line with the concept that NTA of cytosolic NatA substrates prevents the creation of IBM-like N-termini in the cytosol, depletion of NatA activity triggers apoptosis in humans[36]. If apoptosis exists in plants is still controversially discussed[37], and direct homologs of IAPs are missing in plants. The only IAP-like protein (AtILP) of Arabidopsis also contains a RING-domain, but lacks the baculovirus IAP repeat (BIR) domain, which is essential for binding of non-acetylated protein N-termini and anti-apoptotic activity in the human IAP proteins[19,38]. The latter suggests that IAPs are not acting as N-recognins in plants. However, the concept that the N-terminus and C-terminus of proteins are hotspots for determining protein stability is currently emerging in eukaryotes[39]. Similarly, to the nonAc-X²/N-degron, the quality control of N-myristoylated proteins is mediated by a nonmodified glycine at their N-terminus. The non-modified and consequently mislocalized proteins were targeted for proteasomal degradation by two Cul2 containing Cullin-RING E3 ubiquitin ligase complexes through the substrate adaptors ZYG11B and ZER1. If CRLs also contribute to the identification of non-modified X²/N-degrons is currently unclear in plants. However, it is tempting to speculate it since the neddylation of Cul1 in amiNAA10 plants was substantially enhanced. However, clear homologs for the UB E3-ligases ZYG11B and ZER1 are absent in Arabidopsis.

Our findings define the nonAc-X²/N-degron-mediated degradation as a novel hormone-regulated branch of the N-degron pathways in plants targeting a vast number of long-lived cytosolic proteins (ref. [29] and Supplementary Fig. 8). The previously established Arg/N-degron pathway predominantly targets short-lived regulatory proteins, whose N-degrons are conditionally generated by post-translational processing in plants (e.g., by Cys-oxidation or internal cleavage)[40–42]. Thus, the nonAc-X²/N-degron pathway and the Arg/N-degron pathway address different types of protein subsets, causing a potentially different impact on bulk protein turnover. Unlike mutants affected in masking the nonAc-X²/N-degron pathways, loss-of-Arg/N-degron pathway

mutants grow like the wild type plants under non-stressed conditions[35,36]. We conclude from our results that proteostasis of a large number of cytosolic NatA substrates is substantially affected by a tightly controlled ribosome-associated protein modifier that is essential in Arabidopsis and humans and determines the half-lifetime of proteins when they are synthesized.

## Methods

**Plant material and growth.** All analyzed plants belong to the *A. thaliana* ecotype Col-0. The here analyzed NatA depleted mutants amiNAA10, amiNAA15, and *muse6* were generated and characterized previously[9,10]. Plants were germinated and grown on soil (Tonsubstrat (Ökohum®, Herbertingen)) supplemented with 10% (v/v) vermiculite and 2% (v/v) quartz sand. Seeds were stratified for two days on humid soil at 4 °C in the dark and then transferred to short-day conditions (8.5 h day, light intensity: 70–100 μmol m$^{-2}$ s$^{-1}$, day temperature: 22 °C, night temperature: 18 °C, relative humidity: 50–60%) in plant growth chambers (Waiss, Germany). Plants were kept under short-day conditions until indicated time point for time-resolved quantification of protein degradation, steady-state proteome analyses and enrichment of ubiquitinated proteins or translated proteins. For seed production, eight-week-old soil-grown plants were transferred to long-day conditions (16 h light per day, other conditions were the same as in short day cultivation) and grown until seeds matured in the siliques.

**Quantification of global protein turnover by isotope labeling.** The global protein turnover rates in leaves of approximately 6-week-old soil-grown plants were quantified by specific isotope labeling with the EasyTag™ EXPRESS³⁵S protein labeling mix (11 mCi/ml, PerkinElmer).

For determination of translation rates *in planta*, leaf discs (4 mm) obtained from the most mature (largest) leaves of the wild type or NatA depleted plants were floated on ½ Hoagland medium supplemented with 70 μCi/ml for up to 90 min. Samples were harvested at indicated time points, washed, and immediately snap-frozen in liquid nitrogen. The material was ground to a fine powder and extracted on ice with 0.3 ml 50 mM Hepes; 10 mM KCl; 1 mM EDTA; 1 mM EGTA; 10% (v/v) glycerin; pH 7.4 for 15 min. Samples were centrifuged (15,000 g, 4 °C, 15 min), and the supernatant was diluted in 0.3 ml buffer. Excess Easy Tag™ label was removed from the sample (0.15 ml) with a PD Spintrap™ G-25 column (GE Healthcare). An aliquot (30 μl) of the purified protein sample was dissolved in 10 ml liquid scintillation cocktail (Ultima Gold™, PerkinElmer), and the protein-incorporated isotope label was quantified with the Tri-Carb 2810TR Liquid Scintillation Analyser (PerkinElmer). The detected signal was normalized to the fresh weight of the sample after background subtraction.

To quantify the in vivo protein degradation rates, proteins were labeled by incubating leaf discs for 17.5 h with ½ Hoagland medium supplemented with 88 μCi of EasyTag™ EXPRES ³⁵S-protein labeling. At time point zero, the leaf discs (4 mm) were transferred to ½ Hoagland medium containing 1 mM cycloheximide for up to six hours. The cycloheximide containing solution was replaced every 60 min to ensure translation arrest. Samples were harvested at indicated time points, and the proteins were prepared as described above for quantification of

protein-incorporated isotope label. The normalized signal intensity at time point zero was set to 100%.

**Verification of isotope-labeled amino acids incorporation into proteins**. The isotope-labeled proteins were separated by SDS-PAGE, transferred on a nitro-cellulose membrane and visualized on an X-ray film. The exposure time was 48 h.

**Determination of protease activity**. Wild type and NatA depleted mutants were grown for five weeks on soil under short-day conditions. Leaves were harvested, snap-frozen, and ground in liquid nitrogen. Soluble proteins were extracted with protein extraction buffer (50 mM Hepes; 10 mM KCl; 1 mM EDTA; 1 mM EGTA; 10% (v/v) glycerin; pH 7.4). Protease activity of total soluble proteins from five-week-old wild type and NatA depleted mutants was quantified using the EnzChek$^{TM}$ protease assay kit (Invitrogen) following the manufacture's protocol.

**Quantification of proteasome activity in plants**. Soluble proteins were extracted from leaf material, as stated above, with the exception that DTT and PMSF were avoided in the extraction buffer. The soluble proteins (0.1 mg) were dissolved in 100 mM Tris-acetate buffer, pH 7 (0.89 ml) containing 25 μM proteasome substrate I (Z-Leu-Leu-Leu-AMC; Sigma-Aldrich) and incubated at 37 °C for 45 min. The reaction was stopped by the addition of 0.1 ml 10% SDS followed by dilution in 0.9 ml 0.1 M Tris-acetate buffer, pH 9. Cleavage of proteasome substrate I was quantified after excitation (380 nm) of the fluorescent cleavage product at 440 nm with a FLUOstar OPTIMA plate reader (BMG Labtech).

**Determination of total protein content**. Total protein content was quantified in leaves of six-weeks-old wild type and NatA depleted plants. Proteins were extracted in 50 mM Hepes pH 7.4, 10 mM KCl, 1 mM EDTA, 1 mM EGTA, 10% (v/v) glycerin, cOmplete protease inhibitor cocktail (roche). Protein concentration was determined using Roti®-Quant (Carl Roth) and the FLUOstar OPTIMA plate reader (BMG Labtech) following the manufactures protocol.

**SDS-PAGE and quantification of proteins after staining with coomassie or silver**. Extracted proteins were supplemented with 5x-SDS sample buffer (10% SDS, 20% Glycerol, 100 mM Tris pH 7, 0.1% bromophenol blue; 25% 2-mercaptoethanol) and denatured for 10 min at 95 °C. Subsequently, the proteins were separated by SDS-PAGE[43]. The polyacrylamide gels were stained for 10 min in FastGene® Q-Stain while shaking and afterward rinsed two times with H$_2$O. PlusOne Silver Staining Kit (Ge Healthcare) was used to detect low abundant proteins. The resulting signals were recorded with the ImageQuant LAS 4000 and quantified with the ImageQuant TL (GE Healthcare) or the Fiji (http://fiji.sc) image processing applications.

**Quantification of plant proteins by immunological detection**. For immunological detection SDS-PAGE-separated leaf proteins were blotted onto a PVDF membrane that was blocked with 5% BSA in TBS-T (50 mM Tris pH 7.6, 150 mM NaCl, 0.1% Tween-20) or with Western Blocking Reagent (Roche), and subsequently decorated with the primary antibody (dilutions can be found in the respective method section). The primary antibody was detected with goat anti-rabbit IgG (H&L) horse-radish peroxidase conjugate (1:25000, AS09602, Agrisera). The horse-radish peroxidase was visualized with the WesternBright Sirius™ Kit (Biozym) according to the manufacturer's instructions. Signals were recorded with the ImageQuant LAS 4000 v1.2 (GE Healthcare) and quantified with the Image-Quant TL v8 (GE Healthcare). Local background normalization with the rolling ball algorithm implemented in the IMAGE QUANT TL v8 software package (GE Healthcare) was applied for individual signals. Detection of signals was in the dynamic range of the LAS4000 detector according to internal controls performed by the IMAGE QUANT LAS4000 v1.2 software (GE-Healthcare)

**Quantification of polyubiquitinated proteins by immunological detection**. Leaf discs (4 mm) obtained from the most mature (largest) leaves of six-week-old wild type or NatA depleted plants were floated on ½ Hoagland medium supplemented with 50 μM MG132 (Santa-Cruz Biotechnology) in the light at 22 °C for 6 h while shaking at 80 rpm to inhibit proteasome activity. Leaf discs floated on ½ Hoagland supplemented with 0.1% DMSO served as non-inhibited proteasome control. Excess liquid was removed, and the sample was immediately snap-frozen in liquid nitrogen. Proteins were separated on a 10% polyacrylamide gel by SDS-PAGE and subsequently blotted on a PVDF membrane. Polyubiquitinated proteins were detected using α-UBQ11 (1:5000, AS08307, Agrisera).

To detect ubiquitinated OAS-TL A by co-immunoprecipitation, leaf discs (4 mm) of six-week-old plants were floated on ½ Hoagland medium supplemented with 50 μM MG132 (Santa-Cruz Biotechnology) for 6 h. Subsequently, total proteins were extracted with extraction buffer (50 mM Hepes pH 7.4, 10 mM KCl, 1 mM EDTA, 1 mM EGTA, 10% glycerin, 0.1% Nonidet P-40, Abcam, and 1× complete protease inhibitor cocktail, Roche), then equal amount of soluble proteins were incubated with OAS-TL A antisera in the presence of MG132. After gentle shaking for 2 h, 30 μL protein A/G agarose beads (Thermo Fisher) were incubated with the samples for 4 h. The beads were washed with extraction buffer five times

and the specifically bound proteins were eluted with 30 μl of 2× SDS buffer. The eluted proteins were separated on a 12.5% (w/v) SDS-PAGE gel and transferred to an Immobilon®-P PVDF membrane (Carl-Roth) and decorated with OAS-TL A antisera (dilution 1:5,000) or mono- and poly-ubiquitination antibody (FK2, HRP conjugate, Enzo life Sciences, dilution 1:5,000). The signals were detected as described above.

**Quantification of ribosomal RNA**. Total RNA was extracted from leaf tissue from wild-type, NAA10, and NAA15 depleted mutants grown for 7 weeks on soil under short-day using the peqGOLD Total RNA Kits (Peqlab) according to the manufacturer's protocol. Total RNA was prepared using Trizol (Gibco) followed by additional purification using the RNeasy Mini Kit (Qiagen). rRNA was quantified by capillary electrophoresis on an Agilent 2,100 bioanalyzer (Agilent).

**Enrichment of ubiquitinated proteins with UbiQapture-Q matrix**. Proteasome and deubiquitination activity were inhibited in the most mature (largest) leaves of wild type and amiNAA10 plants by floating leaf discs (4 mm) on ½ × Hoagland medium supplemented with 50 μM of MG132 (Santa-Cruz Biotechnology) and 20 μM deubiquitinase inhibitor PR-619 (Sigma Aldrich) for 5 h. After incubation, samples were frozen in liquid nitrogen. Proteins were extracted in 20 mM sodium phosphate buffer pH 7.2, 1% Tween 20, 10 mM DTT, 0.5 mM PMSF, cOmplete protease inhibitor cocktail$^{TM}$ (Roche), and 20 μM PR-619. The resulting crude extract was clarified by centrifugation (30 min at 4 °C, 15000 g) and incubated with 40 μl beads (Ubi-Qapture Q matrix, Enzo life Sciences) overnight. The proteins were eluted from the matrix according to manufacturer's instructions. Same volume of the different elution fractions was separated by SDS-PAGE and visualized by silver staining (GE Healthcare). The immunological detection was done with the antibody mono- and polyubiquitinylated conjugated to HRP (FK2, HRP conjugate, Enzo life Sciences) provided with the Ubi-Qapture Q kit.

**Quantification of neddylated Cullin1**. To quantify the neddylation of Cullin1, total protein was extracted from 6-week-old wild type and NatA depleted plants using 50 mM Tris-HCl pH 7.5, 150 mM NaCl, 10 mM EDTAS, 100 μM PMSF, 10 mM N-ethylmaleimide, 0.1% IGEPAL and cOmplete protease inhibitor cocktail (Roche) in a 1:4 ratio (w/v). The resulting protein crude extract was supplemented with 2x SDS Sample Buffer (125 mM Tris-HCl pH 6.8, 5% 2-mercaptoethanol, 10% glycerol, 2% SDS, 0.01% bromphenol blue). Proteins were subsequently separated by SDS-PAGE (7.5%) and transferred on a pvdf membrane to visualize the neddylated (NEDD8-CUL1, upper signal) and unneddylated (lower signal) variant of CUL1 (anti-CUL1, 1:1000[44]).

**Determination of TOR activity**. TOR activity was quantified in five-week-old plants, total proteins were extracted in 0.25 M Tris-HCl; 8% SDS; 20% glycerol; 0.002% bromephenol blue; 5% 2-mercaptoethanol, pH 6.8 supplemented with 3% phosphatase inhibitor cocktail 2 (Sigma-Aldrich) and cOmplete protease inhibitor cocktail (Roche) with a ratio 1 to 5 (w/v). After separation by SDS-Page and transfer to a pvdf membrane anti-S6k-p (Cell Signaling, 9205, 1:5000), anti-S6K1/2 (1:5000, AS121855, Agrisera) were used to quantify the phosphorylated variant (S6k-P) and the total S6K1/2 content.

**Time resolved destabilization assays of selected protein**. Protein synthesis was inhibited by floating leaf-discs (3 mm) from the most mature (largest) leaves of six-week-old plants grown on ½ × Hoagland medium supplemented with 1 mM of CHX (Sigma Aldrich) for up to 6 h while shaking at 80 rpm. 0.1% v/v of the solvent in ½ × Hoagland medium was used as a control. After the incubation excess liquid was removed from the discs, and the sample was immediately snap-frozen in liquid nitrogen and stored at −80 °C. Soluble proteins were extracted in 50 mM Hepes pH7.4, 10 mM KCl, 1 mM EDTA, 1 mM EGTA, 10% (v/v) glycerin, and cOmplete protease inhibitor cocktail$^{TM}$ (Roche). Same volume of the protein crude extracts was used for separation by SDS-PAGE. Immunological detection was carried out using the following primary antibody and dilutions: α-O-Acetylserine(thiol)lyase A (OAS TL-A, 1:5000[45], α-Glutathione Reductase 1 (GR1, 1:2500[46], α-Regulatory particle subunit 10 (RPN10, 1:2000, AS01012, Agrisera), α-tubulin (1:5000, AS10681, Agrisera), α-Coronatine Insensitive Protein 1 (COI, 1:1000, AS122637, Agrisera), and α-Serine acetyltransferase 5 (SAT5,1:5000[47].

**Enrichment of newly translated proteins by BONCAT**. Newly translated proteins were labeled by floating leaf discs (4 mm) from the most mature (largest) leaves of 6-week-old wild type and amiNAA10 plants for 4 h on ½ Hoagland medium supplemented with 50 μM L-homopropargylglycine (HPG, Invitrogen, for biotin labeling of nascent proteins) or 50 μM L-azidohomoalanine (AHA, Invitrogen, for nascent protein enrichment). After the incubation medium was removed from the leaf discs and subsequently frozen in liquid nitrogen and stored at −80 °C.

To visualize the newly translated proteins, HPG labeled proteins were clicked to Biotin Azide (Invitrogen) using the Click-iT™ Protein Reaction Buffer Kit (Invitrogen) following the manufactures protocol. The biotin-labeled proteins were immunologically detected after separation of the protein crude extract by

SDS-PAGE and blotting onto a PVDF membrane using Neutravidin-HRP (1:100,000, Invitrogen, 31001).

Nascent proteins were enriched by cycloaddition of AHA-labeled proteins to the alkyne agarose resin using the Click-iT™ Protein Enrichment Kit (Invitrogen) following the manufacturers protocol. Proteins were eluted by trypsin digestion from the resin after stringent washing with 1% SDS (5 times), 8 M Urea/100 mM Tris pH 8 (10 times) and 20% acetonitrile (10 times) to eliminate non-specifically bound proteins from the resin.

**Quantitative proteomics of the ubiquitome and the translatome.** Peptides were analyzed as described in[48] with minor modifications. In brief, after overnight trypsin digestion the supernatant was acidified, desalted using stage tips[49] and dried in a vacuum concentrator before nanoLC–MS analysis with an Ultimate 3000 liquid chromatography system coupled to a QExactive HF mass spectrometer (Thermo-Fischer, Bremen, Germany). Samples were dissolved in 0.1% TFA, injected to a self-packed analytical column (75 µm × 200 mm; ReproSil Pur 120 C18-AQ; Dr Maisch GmbH) and eluted with a flow rate of 300 nl/min in an acetonitrile-gradient (3–40%). The mass spectrometer was operated in data-dependent acquisition mode, automatically switching between MS and MS². Collision induced dissociation MS² spectra were generated for up to 20 precursors with normalized collision energy of 29%.

Raw files were processed using MaxQuant version 1.5.3.30[50] for peptide identification and quantification. MS² spectra were searched against the Uniprot *Arabidopsis thaliana* proteome database and the contaminants database by Andromeda search engine with the following parameters: Carbamidomethylation of cysteine residues and Acetyl (Protein N-term), Oxidation (M) as variable modifications, trypsin/P as the proteolytic enzyme with up to 2 missed cleavages was allowed. The maximum false discovery rate for proteins and peptides was 0.01 and a minimum peptide length of 7 amino acids was required. All other parameters were default parameters of MaxQuant. LFQ values were calculated by MaxQuant and used for quantitative data analysis.

**Global proteomics for quantification of protein steady state levels.** Proteins were extracted from leaves of 6-week-old soil grown plants according to the in-StageTip protocol[51] and analyzed on an EASY-nLC 1200 UPLC-system (Thermo Fisher Scientific) that was coupled to a Q Exactive HFX Orbitrap instrument. The resulting MS raw files were analyzed using the MaxQuant software, version 1.6.1.13[50], and peptide lists were searched against the species level UniProt FASTA database. A contaminant database generated by the Andromeda search engine[52] was configured with cysteine carbamidomethylation as a fixed modification and N-terminal acetylation and methionine oxidation as variable modifications. We set the false discovery rate (FDR) to 0.01 for protein and peptide levels with a minimum length of seven amino acids for peptides. The FDR was determined by searching a reverse database. Enzyme specificity was set as C-terminal to arginine and lysine as expected using trypsin and LysC as proteases. A maximum of two missed cleavages was allowed. Peptide identification was performed in Andromeda with an initial precursor mass deviation up to 7 ppm and a fragment mass deviation of 20 ppm. All proteins and peptides matching to the reversed database were filtered out. Bioinformatic analyses were performed using Perseus[53].

**Non-invasive relative protein lifetime measurements in plants.** Protein lifetime was quantified *in planta* with the tandem Fluorescence Timer (tFT) approach, we exchanged the SAT5 sequence in the previously described pBinAR::SAT5-tFT construct[26] with the orf encoding the protein of interest. The required endonuclease restriction sites were fused to these orfs by PCR-amplification with specific primers (Supplementary Table 2). Protein lifetime was quantified *in planta* with the tandem Fluorescence Timer (tFT) approach. For this purpose, the SAT5 sequence in the previously described pBinAR::SAT5-tFT construct[26] was exchanged with the orf encoding the protein of interest (X). The required endonuclease restriction sites were fused to these orfs by PCR-amplification with specific primers (Supplementary Table 9). Arabidopsis leaf cells were transiently transformed with *A. tumefaciens* GV3101 carrying a pBinAR::X-tFT construct. For this purpose, a bacterial culture with an OD₆₀₀ of 0.3 in YEB medium (0.5% (w/v) Bacto™ Tryptone, 0.1% (w/v) Bacto™ Yeast Extract, 0.5% (w/v) beef extract powder, 0.5% (w/v) sucrose, 41.5 mM MgSO₄; pH 7.2) supplemented with 100 µM acetosyringone (Sigma-Aldrich) and the antibiotics rifampicin (50 µg ml⁻¹), gentamycin (25 µg ml⁻¹) and kanamycin (50 µg ml⁻¹) was left to incubate at 28 °C. When the OD₆₀₀ of the culture had reached 0.6, the bacteria were harvested by centrifugation at 5,000 g for 5 min. Subsequently, the pellet was resuspended in induction medium (7.5 mM (NH₄)₂SO₄, 57.4 mM K₂HPPO₄, 33.0 mM KH₂PO₄, 2 mM sodium citrate, 5.8 mM sucrose, 2 mM MgSO₄, 5% (v/v) glycerol; pH 5.7) supplemented with 200 µM acetosyringone and incubated at 30 °C for four hours. Afterwards, the bacteria were once again harvested by centrifugation at 5,000 g for 5 min. The pellet was washed twice with infiltration medium (11 mM MES, 20 mM MgSO₄; pH 5.7) before being finally resuspended in infiltration medium supplemented with 200 µM acetosyringone to a final OD₆₀₀ of 0.3. This bacterial suspension was used to infiltrate four-week-old Arabidopsis plants. The transiently transformed plants were analyzed four days after infiltration. Leaves were placed on a water-covered slide and analyzed with a Nikon A1 confocal microscope equipped with gallium arsenide phosphide

detectors and solid-state lasers for excitation at 405, 488, and 561 nm. The fluorescence signal was detected at 525/50 nm after excitation at 488 nm (sfGFP) and at 595/50 nm after excitation at 561 nm (mCherry). The laser power of the 488- and 561-nm lasers was set to a 1:3 ratio. Quantification of fluorescence images was performed after applying a Gaussian blur (σ = 1) and background subtraction in ImageJ (v.1.52 h; https://imagej.nih.gov/ij). To produce a ratiometric image on a pixel-by-pixel basis, signal intensities measured for the mCherry channel were divided by the intensities measured for the GFP channel using the image calculator function. mCherry-sfGFP ratios were visualized by applying the ImageJ lookup table "Fire" which changes the grayscale values of an image to false color.

**Verification of NatA activity on selected candidates.** Purified His₆:MBP:*At*NAA10 (17 µg)[10] was mixed with oligopeptide (200 µM; GeneCust), 100 µM [3H]-acetyl-CoA (Hartmann Analytic), and 80 µl acetylation buffer (500 mM Tris-HCl pH 7.5, 10 mM DTT, 8 mM EDTA, 0.25% BSA). After incubation of the reaction at 37 °C for 60 min, peptides were pulled down using 100 µl SP Sepharose (50% in 0.5 M acetic acid, Sigma Aldrich). Sepharose was washed three times with 0.5 M acetic acid and two times with methanol. The acetylation reactions were quantified using a Tri-Carb 2810TR scintillation counter (Perki-nElmer, USA). Synthetic peptides were custom-made (GeneCust, France) with a > 90% purity. All peptides consist of seven unique amino acids at the N-terminus followed by 17 amino acids identical to the ACTH peptide sequence (RWGRPVGRRRRPVRVYP) except that the lysines were replaced by arginines to eliminate any potential KAT activity of NAA10. Peptide sequences can be found in Supplementary Table 3.

**Statistical analysis.** The Holm Sidak one-way ANOVA or the unpaired Student's *t*-test of the SIGMA Plot12 software suite was applied for the detection of statistically significant differences between sample groups. Asterisks (*$p \leq 0.05$, **$p \leq 0.01$ ***$p \leq 0.001$) or letters indicate individual groups identified by the Student's test or pairwise multiple comparisons with the Holm Sidak one-way ANOVA, respectively.

The significance of enriched NatA substrates in the different MS quantifications was calculated using the Easy Fisher Exact Test (https://www.socscistatistics.com) against the Arabidopsis nuclear encoded proteome (The Arabidopsis Information Resource Database). The acetylation frequency of the Arabidopsis proteome was extrapolated using the quantified NTA frequencies determined in[10].

**Gene set enrichment analysis (GSEA).** Gene Set Enrichment Analysis (GSEA) was performed by using DAVID Bioinformatics Resources version number 6.8 (https://david.ncifcrf.gov).

**Reporting summary.** Further information on research design is available in the Nature Research Reporting Summary linked to this article.

## Data availability

Mass-spectrometry based proteomics data determining the ubiquitome, the translatome, and the steady-state levels of leaf-proteins in the wild type and NatA depleted lines are deposited in the ProteomeXchange Consortium (http://proteomecentral.proteomexchange.org) via the PRIDE repository (https://www.ebi.ac.uk/pride/) with the data set identifiers PXD024329, PXD024328 and PXD022122, respectively. Microscopy data were uploaded at IDR (https://idr.openmicroscopy.org/). Source data are provided with this paper.

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

## Acknowledgements

We are grateful for the excellent technical and scientific support on confocal laser scanning microscopy by the Nikon Imaging Center (NIC, COS Heidelberg) and on mass-spectrometry-based identification and quantification of proteins by the Core Facility for Mass Spectrometry & Proteomics (CFMP, ZMBH Heidelberg). The Deutsche For-schungsgemeinschaft funded research at Heidelberg University via the Collaborative Research Centre (1036—Project-ID 201348542—SFB 1036) and individual research grants (WI 3560/1-2, WI 3560/4-1, HE 1848/15-2). F.L.P.R and P.M. were supported by the Heidelberg Biosciences International Graduate School (HBIGS).

## Author contributions

E.L. designed and performed in vivo determination of global protein translation and non-invasive quantification of relative protein half-lifetimes with the tandem-FluorescentTimer. P.M. contributed to the biochemical characterization of NatA depleted plants. F.LF.R performed cycloheximide chase assays and purified polyubiquitinated proteins. T.R. designed the LC–MS/MS-based quantification of the ubiquitome and translatome. L.A. repeated cycloheximide chase assays for selected proteins. X.G. per-formed pulldown of polyubiquitinated OAS-TL. J.M. and M.M. quantified steady-state protein levels in wild type, and NatA depleted plants by LC–MS/MS. G.S. designed the immunological approach for specific labeling of neddylated and non-modified Cul1. R.H. supervised F.L.F.R., acquired funding, and contributed to manuscript writing. M.W. designed the study, acquired funding, and wrote the manuscript.

## Funding

## Competing interests

The authors declare no competing interests.
