## [Peer Review File · Nature Communications]

Cotranslational N-degron masking by acetylation promotes proteome stability in plantsREVIEWER COMMENTS

Reviewer #1 (Remarks to the Author):

The manuscript presents convincing data proposing a role for NatA in protein stability and maintaining proteostasis. The authors start with the observation that NatA depletion in plants leads to a significant increase in overall protein degradation mediated by the proteasome. Thus, lack of N-terminal acetylation leads to protein destabilization. Then, the authors show that the steady-state protein levels from NatA-depleted cells are surprisingly quite comparable to wild type proteomes and show that translation/ribosomal activities are increased counteracting the increased degradation after loss of NatA. At the end, the authors show for a number of NatA substrates starting with alanine, glycine or serine that their half-lives are significantly reduced in NatA mutant lines proposing a nonAc-X2/N-degron pathway.

These findings are of general interest and represent an important contribution beyond state of the art which merit publication in Nature Communications. Conclusions drawn are generally backed by sound experimental data. Overall a well written manuscript which aligns well with the current knowledge in the field. However, I have some points which have to be addressed.

Major concerns/suggestions:

1) “enhanced neddylation of Cullin1 demonstrated that Cullin-RING E3 ligases contributed to the enhanced in vivo ubiquitination activity in NatA depleted plants”. The statement comes a bit out of the blue and is missing context. What was the rationale to look specifically for Cullin 1 E3 ligases? Did the authors check other E3 ligases as well? Furthermore, the increased neddylation and expression of Cullin 1 in amiNAA10 and muse is interesting and indicates that Cullin 1 E3 ligases might act as N-recognins but the data are missing a direct functional or mechanistic link to the substrate(s) to make this conclusion. If the authors want to demonstrate the role of Cullin 1 as N-recognine in this pathway and maintain their conclusion, additional experiments are required to support this. Ubiquitination assays might not be feasible but the authors could knock-down Cullin1 or introduce a Cullin 1 mutated neddylation site in the amiNAA10 or amiNAA15 background and show for one substrate (eventually from Figure 4d) a difference in ubiquitination levels or at least that this knock-down/mutation partly rescues the overall phenotype of increased ubiquitination/degradation.

2) Independent of the fact if Cullin 1 acts as N-recognine or other E3 ligases, could the authors for at least one of the identified non-modified-X2/N-degron substrates in Fig. 4d show increased ubiquitination in ami10 or muse6 background compared to wild type by f. e. immunoprecipitation and immunoblot? Have these proteins been identified and quantified as more ubiquitinated in the MS experiment?

3) The authors define the nonAc-X2/N-degron pathway for three groups: alanine starting, glycine starting and serine starting. Glycines N-terminal acetylation is less frequent and proteins are often N-terminal myristoylated. The authors picked HIR3 which is annotated to be myristoylated in its natural state and not to be acetylated. It has been demonstrated that the omission of N-myristoylation leads to degradation by the Gly/N-degron pathway in humans described by Timms et al. Thus, HIR3 indicates that the Gly/N-degron pathway exists as well in plants. If the authors want to show that a nonAc-G2/N-degron pathway exist or glycine residues fall as well in the non-Ac/N-degron pathway, they have to pick an example of a glycine starting protein which is naturally acetylated by NatA and not a myristoylated one.

4) The ubiquitome data are done on an enrichment of ubiquitinated proteins. The MS identification and quantification of these proteins is not based on peptides containing the GlyGly residues from ubiquitin, therefore missing formal proof of ubiquitination. Since the authors did not perform a background control which proteins might just be sticky to the beads, the identified proteins will not be all ubiquitinated. The data are nevertheless convincing and the increase in ubiquitinated proteins evident, but it should be added bona fide ubiquitinated proteins or similar that this is clear. In line with this, for the translome data my questions are, if the raw files have been searched for peptides containing AHA as direct evidence, because it is not stated? Or have background controls been done if the identifications are not based on AHA?

Minor concerns/suggestions

Abstract: „via a previously undescribed and widely conserved non-Ac/N-degron” Please add here in plants or rephrase undescribed. The existence of a non-modified-A2/N-degron pathway in eukaryotes has been described earlier this year by Mueller et al. This paper demonstrates for a number of alanine starting proteins that omission of their natural modification by NatA exposes short N-degron sequence motifs present in approx. 5% of the human proteome which are recognized by inhibitor of apoptosis proteins with E3 ligase function.

Introduction: The IAP-dependent pathway by Mueller et al. overlaps or might be an example of an N-degron pathway acting on a specific subset of proteins of the proposed global N-degron concept in this manuscript in humans. The Gly/N-degron pathway by Timms et al. acts on Gly residues which omitted their N-terminal myristoylation drawing a clear parallel to the concept of nonAc-X2/N-degron pathways. Both papers are in line with the global findings here. I think this literature is relevant and should be mentioned that in humans non-Ac N-degrons or non-modified N-degrons for specific subsets of proteins exist or are emerging.

Results:

- Page4/Line 5: There is written amiNAA10 lines 18 or 24. In the figures the lines are labelled with 10 and 23. A typo? How much is the NAA15 abundance reduced?

- Page5/second paragraph: For the experiment with the UbiQapture-Q matrix, it should be added that the experiment was performed in presence of MG132 in the text or figure legend. It is written in the methods but without this information the accumulation of ubiquitinated proteins seems counterintuitive after increased proteasome activity.

- Figure 1a: ($p < 0.05$, $n=X$). What does X mean? Is the number for every experiment different? N should be added. There are no * for significant different points or small letters. Or to which data points does the $p < 0.05$ refer to?

- Figure 1b: Here it is the first time that small letters are used to refer to statistical groups. It is not intuitive what the letters statistically mean and it is not explained in the legend. (only mentioned in Extended Figure 2 and material and methods, but as well from there it is not clear to me how the groups are defined). Since small letters are used throughout the following figures it should be added to the legend what a,b,c and d mean here.

- Figure 2b: Please indicate or color GR1, OAS-TL A, COL1 and SAT1 in the vulcanoplot

- Figure 3a: Is amiNAA15 #8 overlying with the other two curves? Because I was not sure to see it.

- page 5/last line: Extended data Fig.4, this has to be Extended data Fig. 5.

- "Comparison of the ubiquitome and translome of NatA depleted plants revealed that 65 proteins...". How many of the 232 proteins which show increased ubiquitination were detected in the in the 406 proteins which show increased translation? And how is the overlap in both data sets overall?

- Fig legend 4a: "(a3, black bar)...is the color code meant? It was not clear to me.

- Page 7: Gly; group 2 semicolon instead of colon

- Extended Data 6: There is missing the control for group 3, serine starting proteins.

- Extended Data 7a: Could the picture a bit zoomed? The colors are difficult to see.

Discussion: "The recent identification of E3 ligases (N-recognins), specifically recognizing pathway in humans." I would say Mueller et al. and Timms et al. demonstrated the existence of two non-modified N-degron pathways for specific subsets of proteins in humans and I think it would be worth to extend the discussion by one or few phrases. It is very interesting that the data in the manuscript demonstrate that non-Ac degrons are a general concept at least in plants and taken together it suggests that non-Ac/N-degron or even broader non-modified/Ac-N-degron pathways using different N-recognins might exist.

Reviewer #2 (Remarks to the Author):

N-terminal acetylation (NTA) is a pervasive co-translational modification that has traditionally been associated with increased protein stability, partly because many abundant proteins bear this modification. However, the discovery in 2010 by Varshavsky and colleagues that Nt-Ac can act as an N-degron in specific contexts called the biological significance of NTA into question. This study by Linster et al directly addresses the role of NTA by analysing plants deficient in NatA, the predominant Nt acetyltransferase in *Arabidopsis thaliana*. A wide selection of orthogonal approaches is used to produce some very interesting novel findings.

The authors demonstrate that depletion of NatA activity led to an increase in protein turnover, mostly of canonical NatA targets, which is in line with the classical view of NTA. Novel findings are as follows: (1) the increased turnover of proteins was associated with increased proteasome activity and abundance, (2) translation was also increased, suggesting the existence of a mechanism to maintain steady-state protein abundance, (3) the increase in translation was controlled by TOR, (3) a new nonAc/N-degron was identified via protein lifetime studies. The authors propose that this mechanism will likely be conserved in other eukaryotes.

The manuscript is very interesting and an important potential contribution to the literature, but I have several questions for the authors:

Much of the data is in the form of quantified immunoblots. It is pleasing to see the full blots presented in the supplementary figures, however, there is no statement regarding how many times each experiment was repeated. Given the partly subjective nature of quantifying immunoblot data, it is essential that several independent blots be conducted to give confidence in the results (these could perhaps be shown in the supplementary data). In particular, Extended data figure 5 seems to be dominated by a strong band in one of the replicates. What steps were taken to ensure linearity of the response when quantifying immunoblots?

Although the study presents some intriguing novel findings, several questions remain unanswered. Whilst these may be beyond the scope of the current manuscript (which is already substantial), at least some of these issues could usefully be discussed:

- What is the mechanism for increased proteasome activity? The authors fail to discuss their interesting result. Are transcripts encoding RPN10 and PBA1 increased in NatA-depleted plants? Or is proteasome turnover reduced? Or something else (e.g. PTM, binding protein)? It is interesting that several proteasome-related GO terms were enriched in the poly-Ub dataset from NatA depleted plants.
- Similarly, the enhanced neddylation of Cullin 1 is an intriguing result that is not pursued or discussed further. Cullin 1 appears to be increased in abundance in NatA deficient plants. What is the significance of these results? Are Cullins proposed as E3 ligases that mediate the degradation of proteins bearing the novel nonAc/N-degron? The discussion refers to E3 human E3 ligases (including Cullins; Timms et al., 2019) that recognize non-acetylated NatA substrates but does not make a specific link
- The text focuses on the identification of NatA substrates in the omics datasets. However, a significant number of other proteins are identified, some potential substrates of other Nats and some not. What is the significance of these proteins?

Other points to address:

The Introduction states that NTA occurs in 80-90% of human and Arabidopsis soluble proteins and later that 40 % of proteins are subjected to N-terminal methionine excision, which precedes NTA. Please could the authors explain this apparent paradox for the less specialist reader?

Page 3: the authors state “impairment of NTA by NatA results in a global destabilization of the proteome in Arabidopsis”. The sentence could be rephrased to read “impairment of NatA-dependent NTA....”, which is more precise. Given the relatively low coverage of the proteomics data, this statement is perhaps too strong. In the shotgun mass spec data set, only 92 of 1238 proteins were significantly affected, and the fold changes are not very high. Is this related to the fact that NatA is knocked down, but some activity remains?

Page 4: “increased proteasome activity in NatA depleted plants was independently confirmed” by the immunological detection of proteasome subunits. This is not strictly true, as increased protein abundance does not necessarily imply increased activity. Please rephrase.

Page 4: “the protein profile of soluble proteins... almost not affected in leaves of NatA depleted plants” - this is true for the gel presented (no obvious changes in the band pattern/intensity) but it could be argued that it is not true, based on the proteomics data. However, it is consistent with the relatively low fold changes in the shotgun data.

Page 5: I found the section on OAS-TL A and GR1 very confusing. This may be an issue with the short paper format but some clarity here is essential to understand what look to be superficially contradictory results. Steady-state level of GR1 is down in the proteomics data NatA-depleted plant; OAS-TL is up but not significantly. This appears to disagree with Fig 2e. In Fig 2c, the degradation rates of both proteins are increased in NatA-depleted plants (CHX chase experiments). This makes sense later in the text where translation of OAS-TL A is significantly enhanced but GR1 translation is not.

The protein lifetime studies are a nice feature of the paper, because lifetime is rarely measured rigorously. However, the text and legends should note that the results presented represent relative lifetime- protein half-lives are not measured. The mCherry-sfGFP fusion is apparently designed for proteins with lifetimes between 10 minutes and 8 hours (Khleminskii et al. 2012). From Fig, 2 the half lives of OAS-TL A and GR1 look to be about 5 hours or more, which is within this range. But if proteins have longer half-lives, this may not be detected by this fluorescent protein pair.

The main text mentions canonical NatA substrates in several places, but this is not explained (although it is in the extended data). It would be helpful to include this information more prominently for readers not familiar with Nat specificity.

Figure 2C- the symbols are too small to be legible. Please also increase the size of symbols in Fig 1A. Fig. 1e should have a size marker.

Many of the figures have a colour scheme that is not helpful to colour-blind readers. In some places, the graphs would be clearer without colour (e.g. Fig 4 would work better with grayscale and the current axis labels)

Extended data- a loading control is shown in several panels. What is this? Is it Rubisco large subunit?

Extended data Fig 2e. Where is the loading control?

Extended data Table 2 mentions translatoe in the legend. Is this correct?

Extended data Table 6 has no statistical analysis (although one is presented in Table 7)

It would be very helpful if TAIR annotations are added to the tables, not just the AGI codes.

Methods- which leaves were sampled in the different experiments? This should be stated very clearly in the different methods sections because Arabidopsis rosettes contain leaves at different developmental stages. The detstabilization and BONCAT assays was done with leaf discs: were these from the most mature (largest) leaves?

Response to reviewers

REVIEWER COMMENTS

Reviewer #1 (Remarks to the Author):

The manuscript presents convincing data proposing a role for NatA in protein stability and maintaining proteostasis. The authors start with the observation that NatA depletion in plants leads to a significant increase in overall protein degradation mediated by the proteasome. Thus, lack of N-terminal acetylation leads to protein destabilization. Then, the authors show that the steady-state protein levels from NatA-depleted cells are surprisingly quite comparable to wild type proteomes and show that translation/ribosomal activities are increased counteracting the increased degradation after loss of NatA. At the end, the authors show for a number of NatA substrates starting with alanine, glycine or serine that their half-lives are significantly reduced in NatA mutant lines proposing a nonAc-X²/N-degron pathway.

These findings are of general interest and represent an important contribution beyond state of the art which merit publication in Nature Communications. Conclusions drawn are generally backed by sound experimental data. Overall a well written manuscript which aligns well with the current knowledge in the field. However, I have some points which have to be addressed.

Major concerns/suggestions:

1) “enhanced neddylation of Cullin1 demonstrated that Cullin-RING E3 ligases contributed to the enhanced in vivo ubiquitination activity in NatA depleted plants”. The statement comes a bit out of the blue and is missing context. What was the rationale to look specifically for Cullin 1 E3 ligases? Did the authors check other E3 ligases as well? Furthermore, the increased neddylation and expression of Cullin 1 in amiNAA10 and muse is interesting and indicates that Cullin 1 E3 ligases might act as N-recognins but the data are missing a direct functional or mechanistic link to the substrate(s) to make this conclusion. If the authors want to demonstrate the role of Cullin 1 as N-recognine in this pathway and maintain their conclusion, additional experiments are required to support this. Ubiquitination assays might not be feasible but the authors could knock-down Cullin1 or introduce a Cullin 1 mutated neddylation site in the amiNAA10 or amiNAA15 background and show for one substrate (eventually form Figure 4d) a difference in ubiquitination levels or at least that this knock-down/mutation partly rescues the overall phenotype of increased ubiquitination/degradation.

Response: We fully agree with the reviewer’s conclusion that we have not provided a direct functional link between Cullin–RING ubiquitin E3 ligase complexes (CRL) and the recognition of the nonAc/N-degron. Since Cullins serve as scaffolds for diverse Cullin–RING ubiquitin E3 ligases (CRL), the rationale of this experiment was not to claim that a specific CRL recognizes the nonAcN-degron but to provide further evidence for the strong activation of the ubiquitin-proteasome system in NatA depleted plants. The here shown enhanced neddylation of Cullin 1 suggests that one or many CRLs containing Cul1 display enhanced activity, which contributes to the observed higher ubiquitination rate in NatA depleted mutants. However, we cannot exclude the possibility that these CRLs do not target “nonAc-X²/N-degron proteins” but other destabilized proteins. Such proteins may have become destabilized because of the many pathways affected in NatA depleted plants. For that reason, we have not concluded that a Cullin1 containing CRL is marking “nonAc-X²/N-degron containing proteins” for degradation. This would be an over-interpretation of the presented data.

2) Independent of the fact if Cullin 1 acts as N-recognine or other E3 ligases, could the authors for at least one of the identified non-modified-X2/N-degron substrates in Fig. 4d show increased ubiquitination in *ami10* or *muse6* background compared to wild type by f. e. immunoprecipitation and immunoblot?...

Response: We followed the reviewer's suggestion and showed enhanced poly-ubiquitination of OAS-TL A in *muse6* after affinity enrichment with a specific antiserum. These novel data are shown as new Supplemental Figure 3e.

...Have these proteins been identified and quantified as more ubiquitinated in the MS experiment?

Response: The ten tested cytosolic NatA substrates shown in Fig. 4d were randomly selected. Six of the confirmed nonAc-X²/N-degron containing proteins (CCT1, CTF2B, HAFL, HSP70, NHO1 and SDR3) were also found to be more poly-ubiquitinated according to the quantitative proteomics of poly-ubiquitinated proteins (Fig 1g, Supplementary Table 2).

3) The authors define the nonAc-X2/N-degron pathway for three groups: alanine starting, glycine starting and serine starting. Glycines N-terminal acetylation is less frequent and proteins are often N-terminal myristoylated. The authors picked HIR3 which is annotated to be myristoylated in its natural state and not to be acetylated. It has been demonstrated that the omission of N-myristoylation leads to degradation by the Gly/N-degron pathway in humans described by Timms et al. Thus, HIR3 indicates that the Gly/N-degron pathway exists as well in plants. If the authors want to show that a nonAc-G2/N-degron pathway exist or glycine residues fall as well in the non-Ac/N-degron pathway, they have to pick an example of a glycine starting protein which is naturally acetylated by NatA and not a myristoylated one.

Response: We agree with the reviewer on his statements on the frequency of N-myristoylation and N-acetylation of protein N-termini starting with a glycine. However, not all proteins are 100% NTAed or N-myristoylated. It is well accepted that N-myristoylation efficiency towards a specific peptide sequence directly correlates with the observed partitioning of N-myristoylated proteins between the cytosol and the membrane ^{1, 2}. Thus, the cytosolic fraction of a protein can be NTAed, while the membrane-associated fraction might be N-myristoylated. Indeed HIR3 was found to be myristoylated in the membrane fraction of Arabidopsis cell suspension cultures ³. In Supplementary Figure 6a, we demonstrated that the native HIR3 N-terminus (AT3G01290) can be acetylated by NatA and that a proline at position 3 inhibits acetylation of the HIR3 protein. The HIR3-tFT protein was exclusively localized in the nucleus of leaf cells in the wild-type and the NatA depleted plants (Supplementary Fig. 7a). These findings strongly suggest that the majority of the HIR3-tFT protein was not myristoylated, since myristoylation targets most proteins to membranes ⁴. However, our data do not contradict myristoylation of the HIR3 N-terminus, which might be condition-specific and target a subfraction of the HIR3 to membranes.

4) The ubiquitome data are done on an enrichment of ubiquitinated proteins. The MS identification and quantification of these proteins is not based on peptides containing the GlyGly residues from ubiquitin, therefore missing formal proof of ubiquitination. Since the authors did not perform a background control which proteins might just be sticky to the beads, the identified proteins will not be all ubiquitinated. The data are nevertheless convincing and the increase in ubiquitinated proteins evident, but it should be added bona fide ubiquitinated proteins or similar that this is clear.

Response: We followed the reviewer's suggestion and added the term bona fide.

In line with this, for the translome data my questions are, if the raw files have been searched for peptides containing AHA as direct evidence, because it is not stated?

Response: Peptides containing AHA are covalently linked to the Click-iT™ Protein Enrichment matrix (Invitrogen) by the highly specific Click-iT™ chemistry. Thus, AHA containing peptides cannot be eluted from the matrix. The covalent attachment of the AHA-containing proteins allowed for extensive washing of the Click-iT™ matrix with five column volumes of 1% SDS, 10 column volumes of 8M urea and 10 column volumes of 20% acetonitrile to remove unspecifically bound proteins. After this rigorous washing, peptides representing the AHA-incorporated proteins were eluted from the matrix by on-column trypsin digestion of AHA-incorporated proteins.

...Or have background controls been done if the identifications are not based on AHA?

Response: In addition to the extensive washing, a background control was performed and allowed the identification of only 28 proteins, which represented a minor fraction (3 %) of the 912 proteins that were reproducibly identified in all three replicates of a genotype. Out of those 28 proteins, 4 were significantly more translated in NatA depleted plants. For three of these signals, the intensities for peptides representing these proteins were at least 5-fold higher than the background. In one case (At5g14780), the background intensity was similar to the signal intensity observed in the wild type sample, but the signal in amiNAA10 was approximately 2.5 fold higher than the background. The remaining proteins also found in the background were either translated with an unchanged (12) or lower (12) rate in amiNAA10 when compared to wild type. Data for the background control intensities are added in Supplementary Table 6.

Minor concerns/suggestions

Abstract: „via a previously undescribed and widely conserved non-Ac/N-degron” Please add here in plants or rephrase undescribed. The existence of a non-modified-A2/N-degron pathway in eukaryotes has been described earlier this year by Mueller et al. This paper demonstrates for a number of alanine starting proteins that omission of their natural modification by NatA exposes short N-degron sequence motifs present in approx. 5% of the human proteome which are recognized by inhibitor of apoptosis proteins with E3 ligase function.

Response: We followed the reviewer’s suggestion and specified that the non-Ac/N-degron is undescribed in plants.

Introduction: The IAP-dependent pathway by Mueller et al. overlaps or might be an example of an N-degron pathway acting on a specific subset of proteins of the proposed global N-degron concept in this manuscript in humans. The Gly/N-degron pathway by Timms et al. acts on Gly residues which omitted their N-terminal myristoylation drawing a clear parallel to the concept of nonAc-X2/N-degron pathways. Both papers are in line with the global findings here. I think this literature is relevant and should be mentioned that in humans non-Ac N-degrons or non-modified N-degrons for specific subsets of proteins exist or are emerging.

Response: We fully agree with the reviewer and have mentioned the emergence of these N-degron pathways in the discussion section of the previous manuscript. In the improved version of the

manuscript, this discussion is extended. Furthermore, we added information about the IAP-dependent pathway for the degradation of non-acetylated proteins at the end of the introduction.

Results:

- Page4/Line 5: There is written amiNAA10 lines 18 or 24. In the figures the lines are labelled with 10 and 23. A typo? How much is the NAA15 abundance reduced?

Response: This typo is corrected. The NAA15 protein steady-state level in amiNAA 10 lines 18 and 23 is decreased to approximately 60% of wild type level (Figure 2b and 2d of Linster et al., 2015). The growth phenotype and protein-degradation rate correlated better with the decrease of NAA10 abundance in NatA activity depleted plants (amiNAA10 and amiNAA15 lines) when compared to the decrease of NAA15 abundance in these lines.

- Page5/second paragraph: For the experiment with the UbiQapture-Q matrix, it should be added that the experiment was performed in presence of MG132 in the text or figure legend. It is written in the methods but without this information the accumulation of ubiquitinated proteins seems counterintuitive after increased proteasome activity.

Response: We followed the reviewer's suggestion.

- Figure 1a: ($p < 0.05$, $n=X$). What does X mean? Is the number for every experiment different? N should be added. There are no * for significant different points or small letters. Or to which data points does the $p < 0.05$ refer to?

Response: "n" corresponds to the number of replicates in the experiment and was identical for all genotypes in this experiment. We apologize for copying a preliminary version of figure caption 1 in the main body text of the previous manuscript version. The number for n is added in the improved version of the manuscript. A statistical analysis has been performed and showed that 35S-cysteine in the protein fraction was significantly lower in all lines when compared to wild type from time points 120 to 360 minutes. We omitted the labeling of these statistical differences since the panel is busy. The data are shown as mean +/- standard error. The standard error of the wild type is shown but covered by the black circle.

- Figure 1b: Here it is the first time that small letters are used to refer to statistical groups. It is not intuitive what the letters statistically mean and it is not explained in the legend. (only mentioned in Extended Figure 2 and material and methods, but as well from there it is not clear to me how the groups are defined). Since small letters are used throughout the following figures it should be added to the legend what a,b,c and d mean here.

Response: Small letters indicate individual groups identified by pairwise multiple comparisons with the Holm Sidak one-way ANOVA ($p < 0.05$). The algorithm of the software defines these individual statistical groups and labels them with the same letter if they are not significantly different. Mean values of sample groups that belong to different statistical groups are labeled with different letters. In panel 1b, wild type and amiNAA15 line #10 are statistically not different. Thus, both mean values belong to the same statistical group named group a. The line amiNAA15 belongs to group b, which is statistically different from group a and group c defined in this panel.

However, a determined mean value of a sample group can belong to several statistical groups due to its standard deviation, like in panel 1c. In this case, the mean value receives two letters (e.g., control of amiNAA10 #23 = b and c). The amiNAA10 #23 control is statistically different from all other controls in this experiment (labeled with a), but not different to the MG132 treated wild type (group b) and amiNAA15 and amiNAA10 #18 (group c).

- Figure 2b: Please indicate or color GR1, OAS-TL A, COL1 and SAT1 in the vulcanoplot

Response: GR1 and OASTL-A are labeled in the improved version of Fig 2b. COI1 (COL1) and SAT5 (SAT1) were not identified in the steady-state proteomics approach.

- Figure 3a: Is amiNAA15 #8 overlying with the other two curves? Because I was not sure to see it.

Response: Yes.

- page 5/last line: Supplementary Fig.4, this has to be Supplementary Fig. 5.

Response: Corrected.

- "Comparison of the ubiquitome and translome of NatA depleted plants revealed that 65 proteins...". How many of the 232 proteins which show increased ubiquitination were detected in the in the 406 proteins which show increased translation?

Response: 65 proteins out of the 232 bona fide more ubiquitinated proteins in NatA depleted plants were also more translated in NatA depleted plants. These 65 proteins represent 30 % of the bona fide more poly-ubiquitinated proteins in NatA depleted plants or 16 % of the more translated proteins in NatA depleted plants. Out of these 232 in amiNAA10 bona fide more ubiquitinated proteins only 3 proteins were significantly less translated in amiNAA10.

.... And how is the overlap in both data sets overall?

Response: The overall overlap between the proteins that were detected by both methods was 410 proteins.

- Fig legend 4a: "(a3, black bar)...is the color code meant? It was not clear to me.

Response: Yes.

- Page 7: Gly; group 2 semicolon instead of colon

Response: Corrected.

- Supplementary 6: There is missing the control for group 3, serine starting proteins.

Response: The control for the proline-induced inhibition of plant NatA activity on peptides starting with Ser (group 3) was previously shown in Linster et al., 2015. Additional negative controls refer to the "no peptide control" and the "MPQP-peptide control" that were performed in parallel (left side of the graph).

- Supplementary 7a: Could the picture a bit zoomed? The colors are difficult to see.

Response: Corrected

Discussion: "The recent identification of E3 ligases (N-recognins), specifically recognizing pathway in humans." I would say Mueller et al. and Timms et al. demonstrated the existence of two non-modified N-degron pathways for specific subsets of proteins in humans and I think it would be worth to extend the discussion by one or few phrases. It is very interesting that the data in the manuscript demonstrate that non-Ac degrons are a general concept at least in plants and taken together it suggests that non-Ac/N-degron or even broader non-modified/Ac-N-degron pathways using different N-recognins might exist.

Response: We followed the reviewer's suggestion to intensify the discussion on the non-modified-X²/N-degron in humans.

Reviewer #2 (Remarks to the Author):

N-terminal acetylation (NTA) is a pervasive co-translational modification that has traditionally been associated with increased protein stability, partly because many abundant proteins bear this modification. However, the discovery in 2010 by Varshavsky and colleagues that Nt-Ac can act as an N-degron in specific contexts called the biological significance of NTA into question. This study by Linster et al directly addresses the role of NTA by analysing plants deficient in NatA, the predominant Nt acetyltransferase in *Arabidopsis thaliana*. A wide selection of orthogonal approaches is used to produce some very interesting novel findings.

The authors demonstrate that depletion of NatA activity led to an increase in protein turnover, mostly of canonical NatA targets, which is in line with the classical view of NTA. Novel findings are as follows: (1) the increased turnover of proteins was associated with increased proteasome activity and abundance, (2) translation was also increased, suggesting the existence of a mechanism to maintain steady-state protein abundance, (3) the increase in translation was controlled by TOR, (3) a new nonAc/N-degron was identified via protein lifetime studies. The authors propose that this mechanism will likely be conserved in other eukaryotes.

The manuscript is very interesting and an important potential contribution to the literature, but I have several questions for the authors:

Much of the data is in the form of quantified immunoblots. It is pleasing to see the full blots presented in the supplementary figures, however, there is no statement regarding how many times each experiment was repeated. Given the partly subjective nature of quantifying immunoblot data, it is essential that several independent blots be conducted to give confidence in the results (these could perhaps be shown in the supplementary data)....

Response: Repetitions of individual blots have been added in the source data file for all experiments as suggested by the editor. These independent and novel data confirm the major claims drawn from quantification of the previous immunoblot (e.g. see source file sheet Figure 1c). The authors want to add at this point that they have tried very hard to support the major claims of this manuscript by at least two independent approaches. For example, we showed enhanced endogenous UPS activity by

quantifying the extractable proteasome activity, immunological detection of proteasome subunits, detection of poly-ubiquitinated proteins, and quantification of the internal degradation after feeding of radioactively-labeled amino acids. Furthermore, many of the here presented data are independently repeated in the follow-up manuscript on HYPK that is available for the reviewer.

...In particular, Supplementary figure 5 seems to be dominated by a strong band in one of the replicates. What steps were taken to ensure linearity of the response when quantifying immunoblots?

Response: We rigorously applied local background subtraction with the rolling ball algorithm implemented in the IMAGE QUANT TLV8 software package (GE Healthcare) for individual signals and ensured that the detected signals were not saturated. In all cases, signal detection was still in the dynamic and linear range of the detector according to internal controls performed by the IMAGE QUANT LAS4000 v1.2 software (GE-Healthcare). In most cases, we confirmed these internal controls by ensuring linearity of the signal of interest upon different exposure times. An independent immunological quantification of S6K-phosphorylation in WT and NatA depleted plants is shown in Figure 3C. Furthermore, we have applied an independent approach to confirm increased TOR activity by demonstrating enhanced rRNA amounts in NatA depleted plants. Transcription of rRNA is under control of TOR-triggered S6K-phosphorylation. The enhanced rRNA levels in NatA depleted plants are supported by an enhanced translation rate in NatA depleted plants.

Although the study presents some intriguing novel findings, several questions remain unanswered. Whilst these may be beyond the scope of the current manuscript (which is already substantial), at least some of these issues could usefully be discussed:

- What is the mechanism for increased proteasome activity? The authors fail to discuss their interesting result. Are transcripts encoding RPN10 and PBA1 increased in NatA-depleted plants? Or is proteasome turnover reduced? Or something else (e.g. PTM, binding protein)? It is interesting that several proteasome-related GO terms were enriched in the poly-Ub dataset from NatA depleted plants.

Response: We agree with the reviewer that the increased in vitro proteasome activity is an interesting finding and contributes to explaining the enhanced in vivo protein turnover in NatA depleted plants. In agreement with the enhanced extractable proteasome activity, we detected a higher abundance of proteasome subunits (e.g. PBA1), suggesting that the enhanced in vitro proteasome activity is at least partially caused by a higher steady-state proteasome level. However, steady-state transcripts levels for RPN10 and PBA1 were not increased according to the global transcriptome analysis or detection by qRT-PCR (novel data, Supplementary Figure 2g). In this context, it is noteworthy that the only significantly upregulated pathway in NatA depleted plants was the ubiquitin-mediated proteolysis (ATH04120, $p = 0.01$, ⁵). This information is added in the result section of the improved version of the manuscript. Consequently, the accumulation of proteasome might be driven by transcriptional induction of other proteasome subunits. Indeed, we found the steady-state protein levels of all detected proteasome subunits (15 subunits, Supplementary Table 4) 10 to 50% enhanced in aminNAA10 when compared to wild type. However, we cannot exclude other mechanisms like post-translational modification of one or several subunits of the proteasome. Thus, we would like to refrain from speculating about such potential mechanisms in the discussion but add the following explanatory sentence at the end of the first chapter of the result section. "Our data strongly suggests that NatA depletion caused faster protein degradation by enhancing the endogenous ubiquitination rate and increasing the capacity of the proteasome as shown by enhanced abundance and activity of the proteasome in soluble protein extracts."

- Similarly, the enhanced neddylation of Cullin 1 is an intriguing result that is not pursued or discussed further. Cullin 1 appears to be increased in abundance in NatA deficient plants. What is the significance of these results? Are Cullins proposed as E3 ligases that mediate the degradation of proteins bearing the novel nonAc/N-degron? The discussion refers to E3 human E3 ligases (including Cullins; Timms et al., 2019) that recognize non-acetylated NatA substrates but does not make a specific link

Response: We followed the reviewers' suggestion to discuss the potential contribution of CRLs to the enhanced protein turnover in NatA depleted plants in more detail. We now discuss that it is tempting to speculate that CRLs contribute to the turnover of nonAc-X²/N-degron also in plants, based on our results and the findings provided by Timms et al., 2019. However, we clearly state that we did not provide significant evidence for a contribution of a CRL in this process. In the results section, we specified that the here observed enhanced neddylation of Cul1 contributes to explaining the enhanced total in vivo ubiquitination rate in amiNAA10. The authors feel that identifying the responsible CRL complex(s) recognizing the nonAx-X²/N-degrons is out of the scope of this manuscript because it would require entirely novel experimental approaches and addresses a subordinated question.

- The text focuses on the identification of NatA substrates in the omics datasets. However, a significant number of other proteins are identified, some potential substrates of other Nats and some not. What is the significance of these proteins?

Response: These proteins might well co-regulated in pathways that are affected by destabilization of cytosolic nonAc-X²/N-degron containing protein. Since the NatA complex potentially addresses 40 % of the proteome, we would like to refrain from a discussion on selected proteins. The authors feel that this will dilute the central message of the manuscript and will destroy the concise style of data presentation. Since we have to compare three proteomics approaches to extract the main message, the authors suggest focusing on the main findings that were confirmed for selected proteins by the tFT-reporter study.

Other points to address:

The Introduction states that NTA occurs in 80-90% of human and Arabidopsis soluble proteins and later that 40 % of proteins are subjected to N-terminal methionine excision, which precedes NTA. Please could the authors explain this apparent paradox for the less specialist reader?

Response: We followed the reviewer's suggestion

Page 3: the authors state "impairment of NTA by NatA results in a global destabilization of the proteome in Arabidopsis". The sentence could be rephrased to read "impairment of NatA-dependent NTA...", which is more precise. Given the relatively low coverage of the proteomics data, this statement is perhaps too strong....

Response: Corrected.

In the shotgun mass spec data set, only 92 of 1238 proteins were significantly affected, and the fold changes are not very high. Is this related to the fact that NatA is knocked down, but some activity remains?

Response: Yes.

Page 4: “increased proteasome activity in NatA depleted plants was independently confirmed” by the immunological detection of proteasome subunits. This is not strictly true, as increased protein abundance does not necessarily imply increased activity. Please rephrase.

Response: We rephrased this statement. “The finding of increased proteasome activity in soluble protein extracts of NatA depleted plants was supported by the immunological detection of the accumulation of the lid and the core subunit of the 26S proteasome RPN10 and PBA1, respectively, in *amiNAA10*...”

Page 4: “the protein profile of soluble proteins... almost not affected in leaves of NatA depleted plants”- this is true for the gel presented (no obvious changes in the band pattern/intensity) but it could be argued that it is not true, based on the proteomics data. However, it is consistent with the relatively low fold changes in the shotgun data.

Response: We rephrased this statement. “Despite the induction of the UPS for degradation of NatA substrates, the total protein content was apparently not affected in leaves of NatA depleted plants (Fig 2a, Supplementary Fig. 3a)”

Page 5: I found the section on OAS-TL A and GR1 very confusing. This may be an issue with the short paper format but some clarity here is essential to understand what look to be superficially contradictory results. Steady-state level of GR1 is down in the proteomics data NatA-depleted plant; OAS-TL is up but not significantly. This appears to disagree with Fig 2e.

Response: This apparent disagreement of the datasets is true for GR1. However, data shown in Fig 2e is derived from leaves that were floated for several hours on the feeding solution. We know from similar experiments that such a treatment results in the formation of reactive oxygen species. This oxidative stress apparently caused up-regulation of GR1, which protects plants against stress-induced over-oxidation of the cytosol^{6,7}. Because of this treatment-induced apparent difference, we confirmed the decrease of GR1 in non-stressed *amiNAA10* plants when compared to wild type (Supplementary Figure 3b). When the leaves of soil-grown wild type and *amiNAA10* plants were immediately harvested, GR1 was less abundant in *amiNAA10* according to both: mass-spectrometric quantification or immunological quantification with specific antisera.

.....In Fig 2c, the degradation rates of both proteins are increased in NatA-depleted plants (CHX chase experiments). This makes sense later in the text where translation of OAS-TL A is significantly enhanced but GR1 translation is not.

The protein lifetime studies are a nice feature of the paper, because lifetime is rarely measured rigorously. However, the text and legends should note that the results presented represent relative lifetime- protein half-lives are not measured.

Response: Corrected.

...The mCherry-sfGFP fusion is apparently designed for proteins with lifetimes between 10 minutes and 8 hours (Khleminskii et al. 2012). From Fig. 2 the half lives of OAS-TL A and GR1 look to be about 5 hours or more, which is within this range. But if proteins have longer half-lives, this may not be detected by this fluorescent protein pair.

Response: This is true. The most stable protein (HIR3-tFT) was detected as a nonAc-X²/N-degron containing protein strongly suggesting that the here analyzed proteins were all in the dynamic range of the applied tFT-fluorescent protein pair. The most instable protein harboring a nonAc-X²/N-degron displayed a mCherry/GFP ratio of 0.6. The two proteins that did not contain a nonAc-X²/N-degron displayed mCherry/GFP ratios of 1 (UGE1) and 0.8 (CYP19), which would have allowed to detect destabilization by absent NTA.

The main text mentions canonical NatA substrates in several places, but this is not explained (although it is in the Supplementary). It would be helpful to include this information more prominently for readers not familiar with Nat specificity.

Response: This information is included in the introduction of the improved manuscript.

Figure 2C- the symbols are too small to be legible. Please also increase the size of symbols in Fig 1A. Fig. 1e should have a size marker.

Response: Corrected. This increase in size will cover in several cases the error bars indicating the standard error. A size marker has been added to the improved version of Fig 1e.

Many of the figures have a colour scheme that is not helpful to colour-blind readers. In some places, the graphs would be clearer without colour (e.g. Fig 4 would work better with grayscale and the current axis labels)

Response: We followed the reviewer's suggestion to apply grayscales in Fig. 4 and amended the color scheme in the other figures accordingly.

Supplementary- a loading control is shown in several panels. What is this? Is it Rubisco large subunit?

Response: In these cases, we have indeed shown Rubisco large subunit. In critical cases, we also quantified TUBB4 by immunological detection and used TUBB4 as a control.

Supplementary Fig 2e. Where is the loading control?

Response: Loading control is added.

Supplementary Table 2 mentions translatoe in the legend. Is this correct?

Response: We thank the reviewer for careful reading. This mistake has been corrected. "translatome" has been replaced by "amount of ubiquitinated proteins in leaves".

Supplementary Table 6 has no statistical analysis (although one is presented in Table 7) It would be very helpful if TAIR annotations are added to the tables, not just the AGI codes.

Response: Data of the statistical analysis and the TAIR annotation were added to Supplementary Table 6 in the improved version of the Supplementary table file.

Methods- which leaves were sampled in the different experiments? This should be stated very clearly in the different methods sections because Arabidopsis rosettes contain leaves at different developmental stages. The detstabilization and BONCAT assays was done with leaf discs: were these from the most mature (largest) leaves?

Response: The reviewer's assumption is correct. We always used the largest leaves from both genotypes to prepare leaf discs, and this information is now clearly stated in the Material and Method section.

References:

1. Traverso JA, *et al.* Roles of N-Terminal Fatty Acid Acylations in Membrane Compartment Partitioning: Arabidopsis h-Type Thioredoxins as a Case Study *The Plant Cell* **25**, 1056-1077 (2013).
2. Pierre M, *et al.* N-myristoylation regulates the SnRK1 pathway in Arabidopsis. *Plant Cell* **19**, 2804-2821 (2007).
3. Majeran W, Le Caer JP, Ponnala L, Meinel T, Giglione C. Targeted Profiling of Arabidopsis thaliana Subproteomes Illuminates Co- and Posttranslationally N-Terminal Myristoylated Proteins. *Plant Cell* **30**, 543-562 (2018).
4. Hemsley PA. The importance of lipid modified proteins in plants. *New Phytol* **205**, 476-489 (2015).
5. Linster E, *et al.* Downregulation of N-terminal acetylation triggers ABA-mediated drought responses in Arabidopsis. *Nat Commun* **6**, 7640 (2015).
6. Marty L, *et al.* The NADPH-dependent thioredoxin system constitutes a functional backup for cytosolic glutathione reductase in Arabidopsis. *Proc Natl Acad Sci U S A* **106**, 9109-9114 (2009).
7. Meyer AJ. The integration of glutathione homeostasis and redox signaling. *Journal of Plant Physiology* **165**, 1390-1403 (2008).

REVIEWERS' COMMENTS

Reviewer #1 (Remarks to the Author):

My concerns and questions have been addressed by the authors and I recommend the manuscript for publication in Nature Communications.

I have a minor point. I commented in the first revision that it should be pointed out /clear when speaking about ubiquitome that the proteins in this experiment do not come from an MS identification of PTM/Ub sites (K-GlyGly) but from an Ub enrichment and protein identifications, because this makes a difference in the level of confidence if really all of those proteins are ubiquitinated. The authors followed my suggestion and added bona fide to the ubiquitinated proteins to point this out. Bona fide is now added throughout the paragraph and when re-reading the manuscript, it felt redundant to me.

For me it would be enough to mention it when the experiment is described for the first time on page 4:...."in 1.6-fold more bona fide poly-ubiquitinated proteins captured" (in addition it is added to the figure). I feel this is indicative enough and afterwards they might refer as well to ubiquitinated proteins. I would like to leave this to the authors if they wanted to omit bona fide in the rest of the paragraph.

Reviewer #2 (Remarks to the Author):

I am mostly satisfied that the authors have addressed my comments on the original version.

I could not find the source file with the repeated immunoblots but trust that these have been done. Please add a statement in the relevant figure legends "representative of x independent experiments"?

Please provide details in the methods section of how ribosomal RNA was quantified.

Response to reviewers' comments:**REVIEWERS' COMMENTS****Reviewer #1** (Remarks to the Author):

My concerns and questions have been addressed by the authors and I recommend the manuscript for publication in Nature Communications.

I have a minor point. I commented in the first revision that it should be pointed out /clear when speaking about ubiquitome that the proteins in this experiment do not come from an MS identification of PTM/Ub sites (K-GlyGly) but from an Ub enrichment and protein identifications, because this makes a difference in the level of confidence if really all of those proteins are ubiquitinated. The authors followed my suggestion and added bona fide to the ubiquitinated proteins to point this out. Bona fide is now added throughout the paragraph and when re-reading the manuscript, it felt redundant to me.

For me it would be enough to mention it when the experiment is described for the first time on page 4:....”in 1.6-fold more bona fide poly-ubiquitinated proteins captured” (in addition it is added to the figure). I feel this is indicative enough and afterwards they might refer as well to ubiquitinated proteins. I would like to leave this to the authors if they wanted to omit bona fide in the rest of the paragraph.

Response: We followed the reviewer’s suggestion and removed the term bona fide after mentioning it on page 4 and in the figure caption.

Reviewer #2 (Remarks to the Author):

I am mostly satisfied that the authors have addressed my comments on the original version.

I could not find the source file with the repeated immunoblots but trust that these have been done. Please add a statement in the relevant figure legends "representative of x independent experiments"?

Response: The improved version of the source file contains the repeated immunoblots. We added a statement on the number of independent experiments in the relevant figure legends.

Please provide details in the methods section of how ribosomal RNA was quantified.

Response: This information has been added.